# Mitigating Off-Policy Bias in Actor-Critic Methods with One-Step Q-learning: A Novel Correction Approach

**Baturay Saglam**                                              *baturay.saglam@yale.edu*
*Department of Electrical Engineering*
*Yale University*

**Dogan C. Cicek**                                              *cicek@ee.bilkent.edu.tr*
**Furkan B. Mutlu**                                        *burak.mutlu@ee.bilkent.edu.tr*
**Suleyman S. Kozat**                                          *kozat@ee.bilkent.edu.tr*
*Department of Electrical and Electronics Engineering*
*Bilkent University*

Reviewed on OpenReview: <https://openreview.net/forum?id=CyjG4ZKCtE>

## Abstract

Compared to on-policy counterparts, off-policy model-free deep reinforcement learning can improve data efficiency by repeatedly using the previously gathered data. However, off-policy learning becomes challenging when the discrepancy between the underlying distributions of the agent's policy and collected data increases. Although the well-studied importance sampling and off-policy policy gradient techniques were proposed to compensate for this discrepancy, they usually require a collection of long trajectories and induce additional problems such as vanishing/exploding gradients or discarding many useful experiences, which eventually increases the computational complexity. Moreover, their generalization to either continuous action domains or policies approximated by deterministic deep neural networks is strictly limited. To overcome these limitations, we introduce a novel policy similarity measure to mitigate the effects of such discrepancy in continuous control. Our method offers an adequate single-step off-policy correction that is applicable to deterministic policy networks. Theoretical and empirical studies demonstrate that it can achieve a "safe" off-policy learning and substantially improve the state-of-the-art by attaining higher returns in fewer steps than the competing methods through an effective schedule of the learning rate in Q-learning and policy optimization.

## 1 Introduction

In continuous action domains, deep reinforcement learning (RL) requires large amounts of data to develop optimal policies and scalable agents. Enhancing data and sampling efficiency often involves using a buffer, known as the experience replay memory (Lin, 1992), to store and repeatedly use agents' experiences for training. The advantage of employing experience replay lies in its off-policy nature, where the stored samples generated by various behavioral policies differ from the target policy being optimized. However, this off-policy learning approach comes with potential pitfalls. Despite agents' past experiences possibly aiding future learning stages, the process neglects the diversity of policies that generated the samples. This off-policy error or bias can lead to sudden policy divergence, especially when combined with function approximation and bootstrapping - a problem emphasized by the deadly triad (Sutton & Barto, 2018, chap. 11). Addressing this off-policy bias commonly involves disregarding transitions produced by behavioral policies uncorrelated to the target policy's distribution. This is known as off-policy correction (Harutyunyan et al., 2016), usually achieved through the importance sampling (IS) operator (Owen, 2013; Hesterberg, 2003). This operator assigns weights to each sample or trajectory within temporal-difference (TD) learning (Sutton, 1988; Precup

et al., 2001) based on the likelihood of the actions generated by the target agent's policy. Consequently, this allows for the weighting of off-policy data when calculating gradients for approximated value functions and policies, thereby refining the overall learning process.

## 1.1 A Review of Prior Off-Policy Correction Methods

IS methods, when applied to TD learning, have demonstrated efficiency in RL, whether used for single transitions (Schmitt et al., 2020) or temporally correlated trajectories (Watkins & Dayan, 1992; Degris et al., 2012; Munos et al., 2016; Harutyunyan et al., 2016; Espeholt et al., 2018). Despite their effectiveness, several challenges are present. For instance, trajectory-based methods can over-treat samples, leading to issues such as unnecessary trajectory terminations, high variance, biased estimates, and gradient instability (Munos et al., 2016; Yu et al., 2018). They can also increase the computational burden and risk of gradient explosion as the environment complexity scales (Fujimoto et al., 2019). Conversely, single-step methods, which were designed for and tested in discrete control, may produce inaccurate importance weights (Schmitt et al., 2020), neglecting policy optimization in weight derivation. Furthermore, they are ill-suited for deterministic policy networks as they typically assume each action has a probability to be selected (Saglam et al., 2022a), thereby leaving deterministic policy networks underexplored.

In continuous action spaces, bootstrapped Q-learning algorithms, based on TD learning methods, are commonly used for learning value functions (Mnih et al., 2013; 2015). Off-policy corrections are typically achieved by leveraging off-policy policy gradient (PG) techniques (Gu et al., 2017; Zhang et al., 2019; Graves et al., 2023) or IS methods that weigh the PG (Humayoo & Cheng, 2019). Despite their utility, these techniques that were built on traditional on-policy deep PG methods often require modification for off-policy learning, implicitly prioritizing PG over off-policy data.

While the available correction techniques in continuous control tackle the off-bias problem from a policy optimization perspective, an alternative approach is to consider the off-policy data—following the practices in discrete action domains (Harutyunyan et al., 2016; Munos et al., 2016; Espeholt et al., 2018; Schmitt et al., 2020). This strategy mitigates the impact of off-policy data on the learning process, and, furthermore, uses full-returns or collected trajectories to accomplish off-policy correction. The assumption here is that a stochastic policy is in place.

In this paper, we identify two main limitations of existing importance sampling methods for off-policy correction: their limited applicability to deterministic policy networks (as we formally show in the following sections) and a tendency to undervalue the importance of the off-policy data. While these methods (i.e., IS in discrete control and off-policy PG) have proven successful, they often focus too much on the PG, which may overlook the significance of off-policy data. We delve deeper into these limitations and discuss how our proposed approach aims to address them in Related Work.

## 1.2 AC-Off-POC: A Novel Approach

To address the identified limitations, we introduce the Actor-Critic Off-Policy Correction (AC-Off-POC) algorithm. AC-Off-POC offers a novel approach, shifting focus to the employment of previously collected off-policy data. This is achieved by adjusting their contributions to the training process through a state-independent weighting strategy. Specifically, we construct a multivariate Gaussian distribution based on the numerical deviations between actions chosen by the behavioral policies and the current policy. Each off-policy sample is then weighted according to the divergence from this Gaussian to a standard unit Gaussian. Our empirical results demonstrate that AC-Off-POC significantly enhances the performance of state-of-the-art models in a variety of tasks, outperforming existing off-policy correction methods in challenging continuous control tasks from OpenAI Gym (Brockman et al., 2016). A comparative analysis of the improvements brought by AC-Off-POC over previous studies is presented in Table 1. In the interest of reproducibility, we have made our source code available in our (currently anonymized) GitHub repository[1].

---

[1] https://github.com/baturaysaglam/AC-Off-POC

Table 1: Properties of the prior studies compared with AC-Off-POC. Note that properties are pertinent to the techniques considered in this study excluding policy search methods.

| Property | IS methods | Off-Policy PG methods | AC-Off-POC |
|---|---|---|---|
| Input type (single- or multi-step) | multi-step | both | single-step |
| Applicable to continuous action spaces? | ✗[1] | ✓ | ✓ |
| Applicable to deterministic policies? | ✗ | ✗[2] | ✓ |
| State-dependent? | ✗ | ✗ | ✓ |
| Independent of the PG used? | ✓ | ✗[2] | ✓ |
| Hyperparameter-free? | ✗[3] | ✗ | ✓ |

[1] Except the study by Humayoo & Cheng (2019)

[2] Since the PG is directly replaced

[3] If variance is reduced

## 2 Related Work

Here, we examine the prior studies regarding either their approach or the target domain of operation. To this end, we thoroughly compare our results with respect to the prior art and detail our contributions.

### 2.1 Off-Policy Correction without Deep Function Approximation

The initial studies in importance sampling and off-policy correction have been done by Precup et al. (2000); Watkins (1989). They focused on eligibility traces and learning from delayed rewards, respectively. This evolved into more complex approaches such as weighted importance sampling with linear function approximators (Mahmood et al., 2014), and multi-step expected return formulations employing Monte Carlo techniques for large state and action spaces (Degris et al., 2012). These developments further motivated variance reduction techniques such as the study of Precup et al. (2000). Recently, Ek et al. (2023) derived an off-policy evaluation scheme under out-of-sample conditions, that is, their approach can assess the performance of a policy using past observational data, where a large portion might be missing. This derivation looks out for practical cases in which reaccess to the training data is impossible due to critical or unethical concerns.

Our work differs from these traditional methods in three key aspects. Firstly, our focus is on deep RL, while these methods are not developed or tested in the deep function approximation setting, which is a more challenging task due to convergence issues. Secondly, our algorithm employs one-step off-policy correction to individual transitions rather than trajectories, a distinction from the works of Precup et al. (2000); Degris et al. (2012). This is significant since trajectory-based methods may lead to unnecessary trajectory terminations, high variance, or biased action probability estimates (Schmitt et al., 2020; Yu et al., 2018). Furthermore, unlike Watkins (1989), we learn a policy from instantaneous, rather than delayed, rewards, which simplifies the learning process and enhances interpretability, efficiency, and real-time adaptation while reducing the dependence on precise reward discounting. Finally, we focus on a learning scheme instead of an evaluation framework that assesses the performance of the agent's policy, in contrast to Ek et al. (2023).

### 2.2 Off-Policy Correction in Discrete Control Tasks

The integration of deep function approximators into RL has led to advancements in algorithms, such as RETRACE (Munos et al., 2016), which employs full returns as in on-policy learning. However, this approach was found to introduce a zero-mean random variable at each step, resulting in increased variance (Schmitt et al., 2020). To tackle this, the V-trace algorithm (Espeholt et al., 2018) was proposed for trading off the increased variance for biased return estimates. Later, LASER (Schmitt et al., 2020) was developed, successfully eliminating V-trace's bias without introducing unbounded variance.

These advancements improved performance in traditional Atari games, yet many of them are not originally proposed nor evaluated in continuous action settings. Conversely, our method is specifically designed with continuous control considerations in mind, demonstrating its versatility. Moreover, our approach employs a one-step off-policy correction on individual transitions, unlike RETRACE and V-trace, which rely on full returns or trajectories. This approach enhances computational efficiency, avoids unnecessary trajectory terminations, and minimizes variance and biased action probability estimates that can occur when multi-step formulation is used (Schmitt et al., 2020; Yu et al., 2018).

### 2.3 Importance Sampling in Continuous Action Spaces

Importance sampling has been relatively underemployed in continuous action domains (Saglam et al., 2022a). Techniques, such as DISC (Han & Sung, 2019), address the bias issue of clipped IS weights by individually clipping each action dimension. Batch Prioritized Experience Replay via KL Divergence (KLPER) (Cicek et al., 2021) prioritizes on-policy samples in actor-critic algorithms by manipulating the structure in sampling from the replay buffer. Lately, Ying et al. (2023) considered the bias in off-policy evaluation caused by the reuse of off-policy samples. The authors call this phenomenon *Reuse Bias*, and alleviate it using their algorithm, Bias-Regularized Importance Sampling (BIRIS). These techniques have produced impressive results in environments that were challenging due to the discrepancy caused by off-policy samples.

However, our study presents several distinct properties from the above IS methods. While the method by Han & Sung (2019) apply IS for on-policy learning to weight transitions or trajectories, our approach employs IS for off-policy learning to rectify off-policy bias. Also, the technique by Cicek et al. (2021) prioritizes on-policy samples as an experience replay sampling algorithm, yet our method is an off-policy correction scheme that cooperates with various experience replay sampling algorithms, as demonstrated by our empirical results. Finally, the BIRIS algorithm addresses the off-policy bias in policy evaluation instead of adjusting the contribution of off-policy samples in optimization. Thus, our study differs from BIRIS in terms of the motivation and approach to the problems caused by off-policy learning.

### 2.4 Off-Policy Correction in Continuous Action Spaces

In PG-based off-policy learning, many methods have addressed the instability of the conventional PG techniques (Humayoo & Cheng, 2019; Graves et al., 2023; Gu et al., 2017; Zhang et al., 2019). In the field of offline RL, Bagatella et al. (2022) proposed an exploration framework in the sparse reward setting to overcome the discrepancy between the offline data that demonstrates an excessively deviating task than the task of interest. Although our work shares similarity with the motivation of the study of Bagatella et al. (2022), we focus on online learning instead of learning from a static dataset. Moreover, our approach centers around adjusting the impact of off-policy samples during the learning process, instead of exploration. Lastly, we consider instantaneous rewards rather than a delayed one, unlike Bagatella et al. (2022).

Concerning online PG methods, which is the central scope of our work, the available methods use off-policy correction techniques for policy gradients. As an example, Relative Off-Policy Actor-Critic (RIS-Off-PAC) (Humayoo & Cheng, 2019) uses a smoothed variant of relative IS with a parameter $\beta \in [0, 1]$ to govern the smoothness. Actor-Critic with Emphatic Weights (ACE) (Graves et al., 2023) applies an emphatic weighting scheme based on the study of Sutton et al. (2016), albeit with a misleading excursion objective. Generalized Off-Policy Actor-Critic (Geoff-PAC) (Zhang et al., 2019) rectifies this issue by employing a counterfactual objective and an emphatic approach for the policy gradient. Moreover, Interpolated Policy Gradient (IPG) (Gu et al., 2017) interpolates on- and off-policy updates for actor-critic methods while satisfying performance bounds. These methods employ multi-step bootstrapped Q-learning and have demonstrated promising results. Thus, we benchmark our proposed approach against these methods in our empirical analysis.

These mentioned studies address the shortcomings of traditional PG methods when used in conjunction with off-policy learning, where it is assumed that the transitions used to compute PG are collected by the current policy, i.e., on-policy. They employ various techniques, e.g., emphatic weighting (Sutton et al., 2016) or importance weights (Humayoo & Cheng, 2019), to adapt the PG to off-policy learning. However, our approach deviates by focusing on mitigating the limitations of off-policy data, rather than modifying the policy gradient. The proposed technique employs traditional PG algorithms in correcting the off-policy

samples, which are inherently on-policy. This aspect of our algorithm consolidates a plug-and-play framework for various PG methods *complementing* the existing off-policy PG approaches.

## 3 Background and Notation

In this section, we delve into the technical underpinnings necessary to articulate the proposed algorithm. We commence with the preliminaries of deep RL, followed by our notation for the off-policy setting. Finally, we formally elucidate why many IS methods are not extendable to deterministic neural network policies.

### 3.1 Deep Reinforcement Learning

At each discrete time step $t$, the agent observes a state $s$, chooses an action $a$, receives a reward $r$, and transitions to a next state $s'$. In fully observable environments, the RL problem is represented by a Markov decision process (MDP), a tuple $(\mathcal{S}, \mathcal{A}, P, \gamma)$ where $\mathcal{S}$ and $\mathcal{A}$ denotes the state and action spaces, respectively, $P$ is the transition dynamics such that $s', r \sim P(s, a)$, and $\gamma$ is the constant discount factor. The *value* is defined as the discounted sum of rewards $R_t = \sum_{i=0}^{\infty} \gamma^i r_{t+i}$, where $\gamma$ prioritizes the short-term rewards.

The goal of an agent is to find a policy $\pi$ that maximizes the expected return: $\mathbb{E}_{s \sim P_\pi, a \sim \pi, r \sim P}[R_0]$, where $P_\pi$ is the distribution over the states visited by policy $\pi$. Policy of an agent is stochastic if it maps states to action probabilities $\pi : \mathcal{S} \to p(\mathcal{A})$ or deterministic if it maps states to a single action $\pi : \mathcal{S} \to \mathcal{A}$. Note that we use the term "deterministic policy" to refer to the policies approximated by the deterministic neural networks throughout the article.

In this study, we consider one-step bootstrapped Q-learning (Watkins & Dayan, 1992). There exists an action-value function (Q-function or critic) corresponding to each policy $\pi$ that represents the expected return while following the policy after taking action $a$ in state $s$, which is computed through the Bellman operator $\mathcal{T}^\pi$ (Bellman, 1957):

$$\mathcal{T}^\pi Q^\pi(s, a) = \mathbb{E}_{r, s' \sim P, a' \sim \pi}[r + \gamma Q^\pi(s', a')].$$

The Bellman operator is a contraction for $\gamma \in [0, 1)$ with unique fixed point $Q^\pi(s, a)$, i.e., the resulting error is bounded by $\gamma$ (Bertsekas & Tsitsiklis, 1996). The optimal action-value function, $Q^*(s, a) = \max_a Q^\pi(s, a)$, is obtained through the greedy actions of the corresponding policy, that is, actions that yield the highest action-value.

In the deep setting of RL, the action-value functions are modeled by deep neural networks $Q_\theta(s, a)$, parameterized by $\theta$, also known as Q-networks. The Q-network is trained by minimizing a loss $J(\theta)$ on the TD error $\delta$, the difference between the network output $Q_\theta(s, a)$ and learning target $y$:

$$
\begin{aligned}
y &:= r + \gamma Q_{\theta'}(s', a'), \\
\delta &:= y - Q_\theta(s, a), \\
J(\theta) &= \mathbb{E}_{s \sim P_\pi, a \sim \pi, r, s' \sim P}\big[|\delta|^2\big],
\end{aligned}
\tag{1}
$$

where $\theta'$ is the target network parameters of the Q-network. A target network is practical approach that provides a fixed objective to optimize the network and ensure stability in the updates. They are usually updated by a small proportion $\tau$ at each step, i.e., $\theta' \leftarrow \tau\theta + (1 - \tau)\theta'$, called soft-update, or periodically to exactly match the behavioral networks called hard-update.

In continuous action spaces, finding the optimal action that maximizes the Q-function, i.e., $\max_a Q^\pi(s, a)$, is not feasible due to the infinite number of actions. To overcome this, actor-critic methods use a separate network, called the actor network (or policy network), that selects actions for the given states. The actor network $\pi_\phi$, parameterized by $\phi$, is optimized by a single-step gradient ascent on the policy gradient. The loss for the deterministic policies $J_{\det}(\phi)$ is based on directly maximizing the Q-value estimate of the Q-network:

$$J_{\det}(\phi) = \mathbb{E}_{s \sim P_\pi}[Q_\theta(s, a)]\big|_{a = \pi_\phi(s)}. \tag{2}$$

To realize a differentiable loss $J_{\mathrm{sto}}(\phi)$, stochastic policies employ the log-derivative trick (Williams, 1992) and scale the logarithm by the expected sum of discounted rewards while following the current policy, predicted

by the Q-network:

$$J_{\text{sto}}(\phi) = \mathbb{E}_{s \sim P_\pi}[\log \pi_\phi(a|s) Q_\theta(s, a)]|_{a \sim \pi_\phi(\cdot|s)}. \tag{3}$$

## 3.2 Off-Policy Learning

The agents employ a *behavioral* policy to accumulate experiences within a replay buffer. These experiences, however, do not fully represent the agent's current policy as they are collected through its older versions, thus termed *off-policy samples*. Conversely, experiences gathered through the current policy are labeled *on-policy samples*. During each update step, the agent selects a batch of transitions from the replay buffer employing a sampling algorithm, which may incorporate both on- and off-policy data:

$$(\mathbf{s}, \mathbf{a}, \mathbf{r}, \mathbf{s}')_{i=1}^{|\mathcal{B}|} \sim \mathcal{R}, \tag{4}$$

where $\mathcal{R}$ and $\mathcal{B}$ are used to denote experience replay and the sampled batch, respectively. We consistently apply boldface to refer to the batch (matrix) of each entity, while non-boldface terms represent vectors (for states and actions) or scalar values (for TD error and rewards).

## 3.3 Importance Sampling in Continuous Control

The current IS methods that incorporate action probabilities to perform off-policy correction demonstrate strong theoretical guarantees and exhibit good performance in practical applications (Espeholt et al., 2018; Schmitt et al., 2020; Munos et al., 2016; Harutyunyan et al., 2016). However, these methods are not well-suited for off-policy actor-critic algorithms that rely on deterministic neural networks to represent their policies. To illustrate, the RETRACE algorithm computes the importance weights of transitions using the following approach:

$$\text{RETRACE}(\tilde{\lambda}) = \tilde{\lambda} \min(1, \frac{\pi(a|s)}{\mu(a|s)}), \tag{5}$$

where $\pi$ and $\mu$ are the current and behavioral policies, respectively, and $\tilde{\lambda}$ is a fixed parameter that specifies the maximum allowed distance between the behavioral and evaluation policy. When considering a deterministic network to represent the behavioral and evaluation policies, as observed in studies by Fujimoto et al. (2018) and Lillicrap et al. (2016), it is important to note that the resulting importance weight will consistently yield a value of 1. This is because both policies functions are many-to-one mappings, where states are deterministically mapped to unique actions with a probability of 1. Consequently, the existing studies on IS are not feasible to be applied to deterministic actor-critic methods, since no weighting would occur in the end.

## 4 Mitigating Off-Policy Bias: A Single-Step Approach

We aim to improve learning efficiency by minimizing off-policy bias arising from divergent behavioral policies. To achieve this, we introduce Actor-Critic Off-Policy Correction (AC-Off-POC), with variants for deterministic and stochastic neural network policies. The AC-Off-POC framework applies Jensen-Shannon divergence to correct the impact of off-policy samples in Q-network and policy optimization, requiring no action probability estimates for deterministic policies. Off-policy bias can be remedied by weighting actor network optimization transitions with similarity weights. This approach respects the MDP assumption, which suggests that collected data originate from the actions generated by behavioral policies. Moreover, TD methods provide direct discrepancy correction between target and behavioral policies through cumulative reward objective-based value functions (Harutyunyan et al., 2016), necessitating weighting in Q-network update transitions as well. To this end, we represent policy similarity on a scale from 0 (uncorrelated) to 1 (identical). Next, we culminate in a comprehensive framework featuring AC-Off-POC for actor-critic with one-step Q-learning, validated by theoretical derivations. We show that AC-Off-POC delivers safe off-policy correction via a bounded $\gamma$-contraction mapping. Key insights are highlighted and discussed throughout.

### 4.1 Deterministic Policies

We delve into the AC-Off-POC for deterministic policies, addressing it in two parts. First, Section 4.1.1 provides a technical description of the algorithm. Subsequently, Section 4.1.2 discusses the rationale behind the design choices made in developing this algorithm.

#### 4.1.1 Methodology

We first assume that the multi-dimensional continuous actions at each step $t$ are samples from a multivariate Gaussian distribution. Let $\lambda$ denote the similarity between the distributions of the behavioral policies that collected the sampled off-policy transitions and the current agent's policy (target or evaluation policy). Then, given a sampled batch of transitions as defined by Equation 4, we can express the deterministic policy loss weighted by the similarity coefficient $\lambda$ as:

$$\tilde{J}_{\det}(\phi) = \frac{\lambda}{|\mathcal{B}|} \sum_{i=1}^{|\mathcal{B}|} Q_\theta(\mathbf{s}_i, \mathbf{a}_i). \tag{6}$$

We refer to the $\lambda$ term as *similarity weights* or *off-policy coefficients* throughout. If the majority of behavioral policies within the sampled batch differ significantly from the current policy, the gradient step on the policy and Q-network parameters will be minimal, as both $\lambda$ and loss are computed over the batch. Extending this and using Equation 6, we define the $\lambda$-scaled TD learning loss as:

$$\tilde{J}(\theta) = \lambda \frac{\|\boldsymbol{\delta}\|_2^2}{|\mathcal{B}|}. \tag{7}$$

To gauge the similarity between the current policy and the policies that enacted the off-policy transitions from sampled batch $\mathcal{B}$, we initially forward pass the states from $\mathcal{B}$ through the actor network corresponding to the current policy:

$$\hat{\mathbf{a}} = \pi_\phi(\mathbf{s}).$$

We now have the batch of current policy's action decisions $\hat{\mathbf{a}}$ on the states from the off-policy transitions, and the batch of action decisions of past policies $\mathbf{a} \in \mathcal{B}$. Let $\dot{\mathbf{a}}$ be the batch of numerical differences in the action decisions:

$$\dot{\mathbf{a}} \coloneqq \mathbf{a} - \hat{\mathbf{a}}.$$

Recalling the multivariate Gaussian assumption on the actions, we then construct a multivariate Gaussian distribution from the action difference batch $\mathcal{N}(\dot{\mu}, \dot{\Sigma})$, where

$$\dot{\mu} = \frac{1}{|\mathcal{B}|} \sum_{i=1}^{|\mathcal{B}|} \dot{\mathbf{a}}_i, \tag{8}$$

$$\dot{\Sigma} = \frac{1}{|\mathcal{B}| - 1} \sum_{i=1}^{|\mathcal{B}|} (\dot{\mathbf{a}}_i - \dot{\mu})^\top (\dot{\mathbf{a}}_i - \dot{\mu}).$$

As an empirical validation, the Gaussian interpolation outlined in the latter equation has been previously employed by Cicek et al. (2021), where it has been proven to enhance overall empirical performance. Notice that the computation of the mean across numerical action deviations, and consequently the covariance matrix, does not depend directly on state information. Rather, it is *state-independent*. This independence arises because the weights are assigned based on actions resulting from the application of the policy to states within the off-policy data. This approach emphasizes the outcomes of actions rather than the states themselves. Next, we define the dissimilarity measure as:

$$\rho = \mathrm{JSD}(\mathcal{N}(\dot{\mu}, \dot{\Sigma}) \| \mathcal{N}(0, \sigma\mathbb{I})), \tag{9}$$

where $\sigma$ is the standard deviation of the exploration noise and $\mathbb{I}$ is the identity matrix. In addition, JSD denotes the Jensen-Shannon divergence, the symmetrized and smoothed version of the Kullback–Leibler (KL)

divergence, defined by

$$M = \frac{1}{2}(P + Q),$$

$$\text{JSD}(P\|Q) = \frac{1}{2}\text{KL}(P\|M) + \frac{1}{2}\text{KL}(Q\|M),$$

where $P$ and $Q$ are random variables.

Naturally, if all transitions in the sampled batch match the distribution under the current policy, we have $\rho = 0$ and $\rho \in (0, \infty)$ otherwise in Equation 9. To project the similarity measure into the interval $[0, 1]$ for bounded weights, a non-linear transformation can be applied:

$$\lambda = e^{-\rho}. \tag{10}$$

Notice that for two identical policies we have $\lambda = 1$, and for completely distinct policies we have $\lambda = 0$, making Equation 10 a similarity measure between two policies.

### 4.1.2 Methodological Justifications and Motivations

**The Gaussian assumption.** As the agent continually learns, each action chosen by a different behavioral policy may originate from a separate Gaussian distribution. Importantly, dimensions within an action vector are often correlated (Todorov et al., 2012; Parberry, 2013), reflecting real-world scenarios where the action space dimensions interact. Suppose a robot attempts to stand from a prone position. Here, each angle in the robot's $n$-dimensional joint-angle action space can influence others, underscoring cross-dimensional correlation. A multivariate Gaussian distribution effectively represents this action vector, with the mean vector denoting the deterministic action chosen by the policy, and the covariance matrix embodying the zero-mean Gaussian-distributed additive exploration noise (Williams, 1992), which is a standard method for introducing exploration in leading deterministic PG algorithms (Hollenstein et al., 2022).

**Using Jensen-Shannon divergence.** There are various divergence options to choose from when quantifying the importance of transitions in Equation 9. For instance, one can employ the Rényi divergence (Rényi, 1961; van Erven & Harremos, 2014) due to its convenient expression for the moments of importance weights. However, we choose to begin with KL divergence, derived from Shannon entropy (Shannon, 1948, chap. 6), which quantifies the information contained in a probability distribution. From the RL perspective, the Shannon entropy of a policy distribution reflects the policy's uncertainty, as actions conditioned on observed states are sampled from this distribution. For instance, an optimal deterministic policy that always selects the same action has low environmental uncertainty (e.g., partial observability, adversarial behavior), reflected in low Shannon entropy. Therefore, the KL divergence between the distributions of two policies indicates their differential environmental uncertainty. A large KL divergence implies that the policies are dissimilar, choosing different actions for each state, suggesting different levels of environmental uncertainty.

Information-theoretical intuition provides a basis for employing KL divergence as an off-policy correction metric (Cicek et al., 2021). However, due to its harsh penalization of significantly differing distributions, a symmetric measure like Jensen-Shannon (JS) divergence becomes a better choice. JS divergence offers a smoothed version of KL divergence and does not penalize distribution differences as severely. Thus, considering the unlikelihood of two behavioral policies of an agent in the same environment being entirely distinct (Sutton & Barto, 2018, chap. 11), JS divergence should be used to gauge the similarity between two behavioral policies during training.

Moreover, we note that rather than directly comparing the action difference with a zero multivariate Gaussian in Equation 9, we compute JSD using a reference matrix of zero-mean Gaussian with a diagonal covariance matrix that contains the standard deviation of the exploration noise in the diagonal entries. Otherwise, policies closer to the current policy that executed a transition with exploration noise could be rejected as the action decisions would numerically deviate from the current policy. Hence, we couple the standard deviation of the exploration noise with the reference matrix to form a parameter-free algorithm.

**Employment of the exponential operator.** The chosen exponential operator provides a continuous link between unbounded similarity values to bounded similarity weights without requiring any quantization, and runs in constant-time, i.e., $\mathcal{O}(1)$. Furthermore, its practical runtime can be reduced using SIMD (Single Instruction Multiple Data) instructions that modern CPUs and compilers offer. SIMD instructions facilitate vector operations, such as computing $e^x$ for each element $x$ of a given vector of dimension $n$. A naive implementation would require $n$ exponentiation operations, leading to a runtime of $\mathcal{O}(n)$. However, SIMD instructions can optimize this operation by allowing the same operation to be executed on multiple data elements in parallel (Buyya et al., 2013; Cardoso et al., 2017; Marinescu, 2023). Thus, the exponential operator also provides the advantage of quick and efficient projection.

Finally, we chose not to include a coefficient to modulate decay (i.e., $e^{-\alpha\rho}$) to keep the algorithm parameter-free. Our empirical results indicate that using $e^{-\rho}$ already achieves satisfactory performance across various environments. However, it is possible to introduce a task-specific modulation coefficient to accelerate learning, if required.

**Analysis of the impact of minibatch size.** The deterministic variant of AC-Off-POC computes the numerical discrepancy between actions of the target and behavioral policies, essentially comparing the distributions under the current and off-policy-transition-executing policies. However, a possible concern lies in the minority of batch transitions executed by policies highly similar to the current one. The average-based similarity weight computation, as per Equation 8, may assign a near-zero fixed weight to these transitions, causing information loss. We address this issue when we introduce the stochastic variant. Additionally, we provide an intuitive analysis on the impact of minibatch size in Observation 1.

**Observation 1.** *Let $|\mathcal{B}|$ and $|\mathcal{R}|$ are the fixed minibatch and experience replay buffer sizes used in training. If $|\mathcal{B}| \to |\mathcal{R}|$, then the mean indices of the sampled transitions will be 0.5 in the expectation, and deterministic AC-Off-POC weights become slightly less than 0.5 due to the non-linear transformation. If $|\mathcal{B}| \to 0$, then the AC-Off-POC weights cannot be accurately estimated. Therefore, very large or small minibatch sizes may result in poor performance.*

## 4.2 Stochastic Policies

The application of AC-Off-POC to stochastic policies is straightforward, given that the policies are represented by probability distributions. While these known probability distributions can be directly used in traditional IS methods via eligibility traces, they often necessitate long trajectories executed by each policy, as demonstrated in prior work. Instead, stochastic AC-Off-POC offers an off-policy correction on randomly selected off-policy transitions.

In this variant, we can forgo assumptions on the specific distribution from which actions are sampled. Thus, we can store the parameters of the policy distributions in the replay buffer. For instance, if an agent's stochastic policy is represented by a Beta distribution $B(\alpha, \beta)$, we can store the parameters $\alpha$ and $\beta$ in the experience replay buffer, sampling these alongside states, actions, and rewards during each update step. Hence, the agent samples a batch of off-policy transitions along with the parameters that define the policy distribution:

$$(\mathbf{s}, \mathbf{a}, \mathbf{r}, \mathbf{s}', \alpha_1, \ldots, \alpha_{|\mathcal{B}|}) \sim \mathcal{R}.$$

In this way, each behavioral policy $\eta_i(\cdot)$ that executed the $i^{\text{th}}$ transition can be represented with the corresponding distribution parameters: $\eta_{\alpha_i}(\cdot)$. Instead of comparing the difference batch with a reference Gaussian, we directly measure the similarity between the distribution under the current policy $\pi(\cdot)$ and $\eta_{\alpha_i}(\cdot)$:

$$\rho_i = \mathrm{JSD}(\pi(\cdot) \| \eta_{\alpha_i}(\cdot)),$$

$$\lambda_i = \min\left[\frac{\pi(\mathbf{a}_i|\mathbf{s}_i)}{\eta_{\alpha_i}(\mathbf{a}_i|\mathbf{s}_i)}, e^{-\rho_i}\right].$$

In the latter equation, we clip the similarity weights to obtain a $\gamma$-contraction mapping around the optimal Q-value, which we examine in the proof of Theorem 3. We then construct a similarity weight vector for the sampled batch of transitions:

$$\boldsymbol{\lambda} := \begin{bmatrix} \lambda_1 & \lambda_2 & \ldots & \lambda_{|\mathcal{B}|} \end{bmatrix}^\top.$$

---

**Algorithm 1** Actor-Critic Off-Policy Correction (AC-Off-POC)

---

1: **Input:** $\pi_\phi, \mathcal{B}$
2: **Output:** $\lambda \vee \boldsymbol{\lambda}$
3: **if** $\pi$ is deterministic **then**
4:     Obtain the minibatch of transitions: $(\mathbf{s}, \mathbf{a}, \mathbf{r}, \mathbf{s}')_{i=1}^{|\mathcal{B}|} \sim \mathcal{R}$
5:     Compute the current policy's action decisions on the sampled states: $\hat{\mathbf{a}} = \pi_\phi(\mathbf{s})$
6:     Compute the numerical action difference batch: $\dot{\mathbf{a}} = \mathbf{a} - \hat{\mathbf{a}}$
7:     Construct the multivariate Gaussian distribution $\mathcal{N}(\dot{\mu}, \dot{\Sigma})$:
$$\dot{\mu} = \tfrac{1}{|\mathcal{B}|}\sum_{i=1}^{|\mathcal{B}|} \dot{\mathbf{a}}_i, \qquad \dot{\Sigma} = \frac{1}{|\mathcal{B}|-1}\sum_{i=1}^{|\mathcal{B}|}(\dot{\mathbf{a}}_i - \dot{\mu})^\top(\dot{\mathbf{a}}_i - \dot{\mu})$$
8:     Compute the similarity coefficient $\lambda$: $\rho = \mathrm{JSD}(\mathcal{N}(\dot{\mu}, \dot{\Sigma}) \| \mathcal{N}(0, \sigma\mathbb{I})) \Rightarrow \lambda = e^{-\rho}$
9: **else**
10:     Obtain the minibatch of transitions containing the policy distribution parameters:
    $(\mathbf{s}, \mathbf{a}, \mathbf{r}, \mathbf{s}', \alpha_1, \ldots, \alpha_{|\mathcal{B}|}) \sim \mathcal{B}$.
11:     Compute the similarity coefficients $\boldsymbol{\lambda} = \begin{bmatrix} \lambda_1 & \lambda_2 & \ldots & \lambda_{|\mathcal{B}|} \end{bmatrix}^\top$:
    $\rho_i = \mathrm{JSD}(\pi(\cdot) \| \eta_{\alpha_i}(\cdot)) \Rightarrow \lambda_i = \min[\frac{\pi(\mathbf{a}_i|\mathbf{s}_i)}{\eta_{\alpha_i}(\mathbf{a}_i|\mathbf{s}_i)}, e^{-\rho_i}]$
12: **end**
13: **return** $\lambda \vee \boldsymbol{\lambda}$

---

Similar to Equations 6 and 7, we can derive the weighted policy and critic loss for stochastic actor-critic:

$$\tilde{J}_{\mathrm{sto}}(\phi) = \frac{1}{\|\boldsymbol{\lambda}\|_1}\sum_{i=1}^{\mathcal{B}} \boldsymbol{\lambda}_i \log \pi_\phi(\mathbf{a}_i|\mathbf{s}_i) Q_\theta(\mathbf{s}_i, \mathbf{a}_i)|_{\mathbf{a}_i \sim \pi_\phi(\cdot|\mathbf{s}_i)}, \tag{11}$$

$$\tilde{J}(\theta) = \frac{\|\boldsymbol{\lambda} \circ \boldsymbol{\delta}\|_2^2}{\|\boldsymbol{\lambda}\|_1}, \tag{12}$$

where $\|\cdot\|_1$ represents the L$_1$ norm and $\circ$ is the Hadamard product.

In the stochastic variant each off-policy transition carries a unique weight, as it can be computed through the corresponding policy distribution. Hence, we compute the loss for actor and critic parameters using a weighted sum of the off-policy transitions, which contrasts the deterministic case. This approach circumvents the potential issue of information loss in the deterministic variant. Nevertheless, even in the deterministic case, such loss of information can be largely disregarded, as the sampled batch is generally expected to be similar to the current policy on average. With this understanding, we extend our discussions on deterministic policies to the stochastic variant of AC-Off-POC. Finally, we provide a pseudocode for AC-Off-POC in Algorithm 1.

### 4.3   AC-Off-POC for Actor-Critic with One-Step Bootstrapped Q-learning

Algorithm 2 elucidates a general framework incorporating AC-Off-POC. Subsequently, we address the potential limitations of AC-Off-POC and its advantages over previous work. We then conduct an intuitive complexity analysis of our method. Lastly, our theoretical analysis, specifically Theorem 3, establishes that our off-policy correction method can generate a $\gamma$-contraction mapping around the optimal Q-value with the TD(0) algorithm (Sutton, 1988). This analysis leverages the tabular setting, routinely used in the deep RL literature to offer theoretical assurances for non-tabular and deep function approximation contexts (Munos et al., 2016; Schmitt et al., 2020; Espeholt et al., 2018; Fujimoto et al., 2018; 2019).

#### 4.3.1   Potential Failure Cases

**Mean difference is zero.**   Even when the behavioral policies in a sampled batch are quite different from the current policy, the computed mean difference could be zero. To see this, suppose that the action range is $[-10, 10]$, and the current policy outputs action 0, where half of the sampled past actions are -10, and

---

**Algorithm 2** A General Framework of AC-Off-POC for Actor-Critic with One-Step Bootstrapped Q-learning Off-Policy

---

1: Initialize the agent with actor $\pi_\phi$ and critic $Q_\theta$ networks with parameters $\phi, \theta$
2: Initialize target and additional networks if required
3: Initialize the experience replay buffer $\mathcal{R}$
4: **for** each exploration time step **do**
5:     Explore the environment and collect a transition tuple $(s, a, r, s')$
6:     **if** the policy is stochastic **then**
7:        Include the policy distribution parameters $\alpha$ to the collected tuple
8:     **end**
9:     Store transition tuple into $\mathcal{R}$
10: **end for**
11: **for** each training iteration **do**
12:     Sample a batch of transitions $\mathcal{B}$ from $\mathcal{R}$ by a sampling algorithm
13:     Obtain the similarity weight(s): $\lambda \vee \boldsymbol{\lambda} = $ **AC-Off-POC**$(\pi_\phi, \mathcal{B})$
14:     Update actor and critic networks with an actor-critic algorithm with one-step bootstrapped Q-learning using the losses weighted by $\lambda \vee \boldsymbol{\lambda}$, e.g., Equations 6, 7, 11, and 12
15:     Update target networks if required
16: **end for**

---

the other half are 10. The fitted normal distribution will have a zero mean even though the policies differ considerably. In this particular example, the variance in the fitted normal distribution would mitigate this issue, that is, the empirical variance would be greater than the exploration noise added to the actions, and thus, the similarity weights will be small.

**Different policies selecting the same action.** Relatively different policies might select the same action for a particular state. Yet, their probability distributions may not similar to each other. Then, we would give low importance weights to the corresponding tuple. However, optimization of stochastic policies gradients is not similar to the deterministic policies since the distribution-related parameters are included in the training. For example, in the Soft Actor-Critic (SAC) algorithm (Haarnoja et al., 2018), (log-) probabilities of the actions rather than only the selected actions and the entropy of the policy distributions are also included in calculating the policy gradient. Hence, the distributions themselves should also contribute to computing the similarities. Thus, even if two policies choose the same actions, we give low similarity weights to the corresponding tuple unless their distributions are also similar.

### 4.3.2 Advantages Over the Prior Works

AC-Off-POC addresses the issues associated with trajectory-based IS, including high variance build-up and vanishing gradients. It avoids these issues by employing individual transitions that are not temporally correlated, with a maximum weight of one. Thus, the weights of the sampled transitions in a single update do not influence the weights in subsequent updates. This makes AC-Off-POC a more favorable choice for off-policy corrections compared to standard IS methods.

Recall that off-policy weighting in TD learning methods is crucial to adjust the expected return under the assumption that off-policy data was generated by the current policy. The existing off-policy correction studies in continuous control focus on off-policy PG. However, these approaches incur high computational complexity, especially with increasing state and action dimensions, as they necessitate lengthy trajectory collection and task-specific hyperparameter tuning. Their adaptability to deterministic policies is also constrained, as they rely on action probabilities, infeasible with deterministic policy approximations via neural networks. An alternative off-policy correction method should also consider off-policy bias in relation to off-policy data, rather than policy gradients, and be compatible with deterministic policies. Our presented solution effectively and safely addresses off-policy correction, both theoretically and practically, as demonstrated through the following rigorous analyses and empirical studies.

### 4.3.3 Complexity Analysis

Primarily, AC-Off-POC necessitates various arithmetic matrix operations and one forward pass through the actor network. The forward pass includes multiple matrix multiplications, linear in input size, overshadowing the complexity of other matrix operations within AC-Off-POC. Thus, the complexity here is $\mathcal{O}(m)$. Conventionally, a simple deep actor-critic algorithm employs two networks for the actor and critic. Backpropagation, similar to the forward pass, is linear in input size, yielding the same complexity if trained over a single iteration. Consequently, each learning step per iteration demands $\mathcal{O}(m) + \mathcal{O}(m+n)$ for the actor and critic networks respectively. Given that the complexity brought by AC-Off-POC is $\mathcal{O}(m)$, it is inferred that AC-Off-POC escalates the complexity of actor-critic algorithms by slightly less than 50% (at most).

### 4.3.4 Theoretical Analysis

In our analysis, we adopt the framework established by Munos et al. (2016). This approach enables us to explain the requirement of clipping the importance weights in AC-Off-POC. Furthermore, by adapting the principles from Munos et al. (2016) to the context of single-step returns on a randomly sampled batch of transitions, we demonstrate that our method functions as a $\gamma$-contraction mapping around the true Q-function $Q^\pi$. Such a property ensures that iterative application of the TD operator moves the estimated value function closer to the true value function, i.e., convergence to the true value function.

**Definition 1.** *AC-Off-POC can be regarded as a special case of the RETRACE algorithm (Munos et al., 2016). Therefore, the general operator for single-step off-policy correction by AC-Off-POC, which is the reduction of RETRACE to a one-step backup setting, is defined by*

$$\mathcal{H}Q(s,a) \coloneqq Q(s,a) + \mathbb{E}_\eta[r + \gamma\mathbb{E}_\pi Q(s',\cdot) - Q(s,a)],$$

*for any behavioral policy $\eta$ that collected the off-policy samples, . . . where we define $\mathbb{E}_\eta[\cdot] \coloneqq \mathbb{E}_{s\sim P_\eta, a\sim\eta, r, s'\sim P}[\cdot]$ and $\mathbb{E}_\pi Q(s,\cdot) \coloneqq \sum_a \pi(a|s)Q(s,a)$, taken over the dataset of interest containing the transitions.*

**Assumption 1** (coverage assumption). *There is a non-zero probability that the current agent's or evaluation policy executes an action with a non-zero probability of being executed under a behavioral policy:*

$$\pi(a|s) > 0 \implies \mu(a|s) > 0, \qquad \forall s \in \mathcal{S}, \quad \forall a \in \mathcal{A}. \tag{13}$$

**Observation 2** (Implication of Assumption 1). *Assumption 5 guarantees that the similarity weights fall within the range $[0, \frac{\pi(y|s')}{\eta(y|s')}]$. This property emerges from our unique computation method, independent of direct action probabilities under the behavioral and target policies. This independence, coupled with the inherent upper bound on the similarity weights, ensures a controlled variance of the IS estimator even when the behavioral policy's action probability is small, thereby contributing to the robustness and efficiency of AC-Off-POC.*

**Lemma 2.** *The difference between $\mathcal{H}Q$ and its fixed point $Q^\pi$ is*

$$\mathcal{H}Q(s,a) - Q^\pi(s,a) = \gamma\mathbb{E}_{s'\sim P_\eta, a'\sim\eta}[\mathbb{E}_\pi(Q - Q^\pi)(s',\cdot) - \lambda'(Q - Q^\pi)(s',a')],$$

*where $\lambda'$ is the similarity coefficient computed for the subsequent transition of $(s,a,r,s')$.*

*Proof.* See Appendix A. □

**Theorem 3.** *If for each action selected by the current policy $a_{i,\pi} \sim \pi$ and sampled batch of transitions $\mathcal{B}_j \sim \mathcal{R}$, we have the similarity coefficient $\lambda_i = \Lambda(a_{i,\pi}, \mathcal{B}_j) \in \min[\frac{\pi(a_{i,\pi}|s_i)}{\eta(a_{i,\pi}|s_i)}, e^{-\rho_i}]$ for the transition tuple $(s,a,r,s')_i \in \mathcal{B}_j$ collected by the behavioral policy $\eta_i$. Then, considering that Assumption 1 is satisfied, for any Q-function $Q$, we have $\mathcal{H}$ being a $\gamma$-contraction mapping around $Q^\pi$:*

$$\|\mathcal{H}Q - Q^\pi\|_\infty \leq \gamma\|Q - Q^\pi\|_\infty,$$

*where $\|\cdot\|_\infty$ is the supremum norm.*

*Proof.* The proof considers stochastic policies and reduction to deterministic policies is straightforward, as we show. From Lemma 2, the difference between $\mathcal{H}Q$ and its fixed point $Q^\pi$ is expressed by

$$
\begin{aligned}
\mathcal{H}Q(s,a) - Q^\pi(s,a) &= \gamma \mathbb{E}_{s' \sim P_\eta, a' \sim \eta} \left[ \mathbb{E}_\pi \Delta Q(s', \cdot) - \lambda' \Delta Q(s', a') \right] \\
&= \gamma \mathbb{E}_{s' \sim P_\eta} \left[ \mathbb{E}_\pi \Delta Q(s', \cdot) - \mathbb{E}_{a'_\pi, s' \sim P_\eta, a' \sim \eta, \mathcal{B}' \sim \mathcal{R}} [\Lambda(a'_\pi, \mathcal{B}') \Delta Q(s', a') | \mathcal{B}'] \right] \\
&= \gamma \mathbb{E}_{s' \sim P_\eta, \mathcal{B}' \sim \mathcal{R}} \Big[ \sum_y (\pi(y|s') - \eta(y|s') \Lambda(y, \mathcal{B}')) \Delta Q(s', y) \Big],
\end{aligned}
$$

where $\mathcal{B}'$ is the batch containing the subsequent transitions of $(s, a, r, s')_{i=1}^{|\mathcal{B}|}$, i.e., $(s', a', r', s'')_{i=1}^{|\mathcal{B}'|} \in \mathcal{B}'$, and $\lambda' = \Lambda(a'_\pi, \mathcal{B}')$ is the corresponding similarity coefficient. Additionally, to obtain non-negative probabilities in the latter equation, $\Lambda(y, \mathcal{B}') \leq \frac{\pi(y|s')}{\eta(y|s')}$ must be satisfied. This is why we clip $\lambda_t = \Lambda(a_t, \mathcal{B}_t)$ for the corresponding pair $(s_t, a_t)$ in the stochastic variant. For deterministic policies, we can have either $\eta(y|s') = 1$ or $\eta(y|s') = 0$. This is because the target and behavioral policies are many-to-one functions, mapping each state with a single action that has a probability of 1. Moreover, since $y$ represents the action chosen by the target deterministic policy, we always have $\pi(y|s') = 1$. Therefore, the constraint for the first case is trivially satisfied for any $\lambda \in [0, 1]$. The constraint for the second case also holds, as can be shown by substituting $\eta(y|s')$ with 0 and following the rest of the proof.

Having $\pi(y|s') - \eta(y|s') \Lambda(y, \mathcal{B}') \geq 0$ satisfied, we have

$$
\mathcal{H}Q(s,a) - Q^\pi(s,a) = \sum_{x,y} w_{x,y} \Delta Q(x, y),
$$

which is a linear combination of $\Delta Q(x, y)$ weighted by non-negative coefficients $w_{x,y}$:

$$
w_{x,y} := \gamma \mathbb{E}_{s' \sim P_\eta, \mathcal{B}' \sim \mathcal{R}} \left[ (\pi(y|s') - \eta(y|s') \Lambda(y, \mathcal{B}')) \mathbb{I}\{s' = x\} \right],
$$

where $\mathbb{I}(\cdot)$ is the indicator function. The sum of those coefficients over $x$ and $y$ is

$$
\begin{aligned}
\sum_{x,y} w_{x,y} &= \gamma \mathbb{E}_{s' \sim P_\eta, \mathcal{B}' \sim \mathcal{R}} \Big[ \sum_y \pi(y|s') - \eta(y|s') \Lambda(y, \mathcal{B}') \Big] \\
&= \gamma \mathbb{E}_{a'_\mu, \mathcal{B}' \sim \mathcal{R}} [1 - \Lambda(a'_\pi, \mathcal{B}') | \mathcal{B}'] \\
&= \mathbb{E}_{a'_\mu, \mathcal{B}' \sim \mathcal{R}} [\gamma - \gamma \Lambda(a'_\pi, \mathcal{B}') | \mathcal{B}'].
\end{aligned}
$$

Clearly, we have $\sum_{x,y} w_{x,y} \leq \gamma$ since $\Lambda(a'_\pi, \mathcal{B}') \leq 1$ for $\forall a'_\pi, \mathcal{B}'$. Therefore, $\mathcal{H}Q(s,a) - Q^\pi(s,a)$ is a sub-convex combination of $\Delta Q(x, y)$ weighted by non-negative coefficients $w_{x,y}$ which sum to at most $\gamma$. Hence, the operator $\mathcal{H}$, which reduces the RETRACE operator to one-step return-based off-policy algorithms with AC-Off-POC, is a $\gamma$-contraction mapping around $Q^\pi$. $\qquad\square$

# 5    Experiments

We conduct experiments to evaluate the effectiveness of the proposed approach. In Section 5.2, we present the simulation results for the comparative evaluation with different IS methods and off-policy PG techniques. We conduct experiments under different batch and replay memory sizes, experience replay sampling methods, and divergence measures in Section 5.3 as a sensitivity analysis. In addition, weights produced by AC-Off-POC are visualized and discussed in Section 5.4. We use the MuJoCo (Todorov et al., 2012) and Box2D (Parberry, 2013) benchmarks interfaced by OpenAI Gym (Brockman et al., 2016). For reproducibility, we did not modify the environment dynamics and reward functions. The computing infrastructure used to produce the reported results is summarized in our repository[1].

## 5.1    Experimental Setup and Implementation

We apply our method to three baseline algorithms: Deep Deterministic Policy Gradient (DDPG) (Lillicrap et al., 2016), Soft Actor-Critic (SAC) (Haarnoja et al., 2018), and Twin Delayed DDPG (TD3) (Fujimoto

et al., 2018). For comparative evaluation, we consider each baseline with and without AC-Off-POC, the continuous IS method, RIS-Off-PAC, and off-policy PG techniques with multi-step bootstrapped Q-learning: ACE, Geoff-PAC, and IPG. In the sensitivity analysis, we use the experience replay sampling methods, Combined Experience Replay (CER) (Zhang & Sutton, 2017), Experience Replay Optimization (ERO) (Zha et al., 2019), and Prioritized Experience Replay (PER) (Schaul et al., 2016). The stochastic variant of AC-Off-POC is applied to SAC, while DDPG and TD3 use deterministic policies.

The environmental parameters used are in accordance with the OpenAI Baselines3 Zoo[2] (Raffin, 2020). Due to the initial exploration steps sampling actions from the environment's action space, policy distribution parameters remain inaccessible for stochastic AC-Off-POC. Therefore, we only use them once they become available after a given number of exploration time steps. In our DDPG implementation, we adhered closely to the parameters outlined in the originating paper. We added an initial 1000 frozen exploration time steps and amplified the batch size to 256 for a more substantial off-policy sample exposure during updates. To facilitate exploration and align with the reference Gaussian distribution, the exploration noise was replaced with a Gaussian with zero mean and standard deviation 0.1. The TD3 implementation uses the finetuned parameter setting found in the author's GitHub repository[3]. The precise implementation details of the baseline algorithms can be accessed via our code[1].

The multi-step PG methods, ACE, Geoff-PAC, and IPG are applied to the baselines by replacing the PG computation. The IS method RIS-Off-PAC is directly applied on top of the baseline algorithms. We follow the original papers to implement ACE, IPG, and RIS-Off-PAC. Our implementation of Geoff-PAC is based on the code from the authors' GitHub repository[4]. We follow the exact structure given in the original papers to implement CER and ERO. Additionally, the repository[5] is used to implement PER.

The competing algorithms introduce hyperparameters that either control the bias-variance trade-off or the on-policyness of the updates. We use the tuned hyperparameters provided in the original articles. In particular, we use the bias-variance trade-off value of $v = 1$ for IPG. As reported by Zhang et al. (2019), ACE is not sensitive for the bias parameter $\lambda_1$ on OpenAI Gym benchmarks and $\lambda_1 = 0$ can produce sufficiently good results. Hence, we set $\lambda_1 = 0$ for ACE. For Geoff-PAC, however, we employ the tuned hyperparameters of $\lambda_1 = 0.7$, $\lambda_2 = 0.6$, and $\hat{\gamma} = 0.2$ since they are reported to result in decent empirical performance. Finally, we use 0.2 for the smoothing control parameter $\beta$ in RIS-Off-PAC.

For all simulations, each algorithm is run for 1 million time steps with evaluations occurring every 1000 time steps, where an evaluation of an agent records the average reward over ten episodes in a distinct evaluation environment without exploration noise and updates. We report the average evaluation return of ten random seeds for initializing networks, simulators, and dependencies. Uniform sampling and an experience replay buffer of size 1 million are used in the comparative evaluations.

## 5.2 Comparative Evaluation

Evaluation results under the DDPG, SAC, and TD3 algorithms are reported in Figures 1, 2, and 3, respectively. As learning curves are intertwined and hard to follow for some of the environments, we also report the average of the last ten evaluation returns, i.e., where the algorithms converge, in Tables 2 and 3. Note that for some of the tasks, the baselines have worse performance than reported in the original articles. This is due to the stochasticity of the simulators and the used random seeds. Nevertheless, the performance difference between competing methods would be consistent if we used different sets of random seeds, regardless of where the baselines converge.

Both variants of AC-Off-POC demonstrate significant performance gains over the baseline algorithms and match or surpass the competing methods across all tested domains. This underscores the scalability of our method. In particularly challenging tasks, such as Ant, BipedalWalker, Humanoid, and Swimmer, where baselines often become trapped at local optima, the benefits of AC-Off-POC are especially evident. The complexities of these environments often originate from the high dimensional state-action spaces or the

---

[2]https://github.com/DLR-RM/rl-baselines3-zoo
[3]https://github.com/sfujim/TD3
[4]https://github.com/ShangtongZhang/DeepRL
[5]https://github.com/sfujim/LAP-PAL

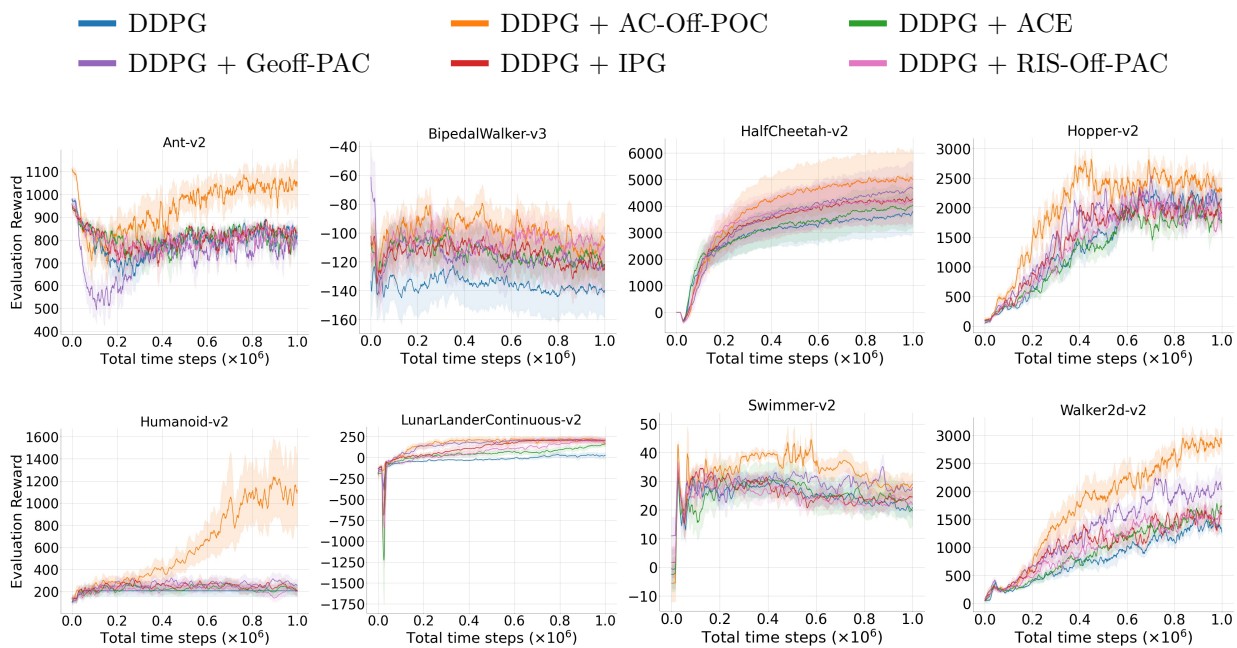

Figure 1: Evaluation curves for the set of OpenAI Gym continuous control tasks for 1 million training time steps over 10 random seeds under the DDPG algorithm. The shaded region represents a 95% confidence interval over the trials. A sliding window of size 10 smoothes the curves for visual clarity.

internal simulation dynamics, which consequently intensify the detrimental effects of off-policy samples. Therefore, the improvements granted by effective off-policy correction techniques become more significant. We note a substantial improvement offered by AC-Off-POC in environments such as BipedalWalker, Hopper, Swimmer, and Walker2d. It has been observed by Henderson et al. (2018) that on-policy algorithms tend to significantly outperform off-policy methods in these tasks. This implies that these environments may heavily rely on on-policy samples for stable and efficient learning. Therefore, our method of correcting off-policy transitions maximizes the improvement, especially in environments where off-policy algorithms often excel, such as Ant, HalfCheetah, and Humanoid. Despite this, we still notice enhanced performance in these domains. This could be attributed to the effective elimination of highly divergent off-policy samples collected throughout the learning process, as the JSD measure only assigns low scores to transitions reflecting very distinct behavioral policies. Although the transitions reflecting different behavioral policies may not be similar, they are still incorporated into Q-learning and policy loss. This insight underscores the discussions regarding the use of JSD and the crucial role of off-policy samples in solving certain environments.

While the enhancements in performance under the SAC and TD3 algorithms remain relatively consistent, a marginally greater improvement is noticed for stochastic policies over deterministic ones when DDPG is included. However, the augmentation of deterministic policy performance becomes pronounced with the application of minibatch learning, as AC-Off-POC entirely discounts transitions within the sampled batch. These results suggests that a more substantial improvement can be expected for stochastic policies. Still, if the distribution of sampled transitions aligns on average with the current policy distribution, the performance can still be enhanced.

Competing methods display significant stability in unstable environments such as Ant, Hopper, and Walker2d. However, the corresponding performance improvements fall short as they directly discard off-policy samples through policy gradients or IS. Detailed examination reveals that Geoff-PAC and IPG yield the best, comparable results among competitors, while RIS-Off-PAC underperforms, and ACE ranks midway. RIS-Off-PAC, using the action-value generated from the behavioral policy to train the algorithm, is typically

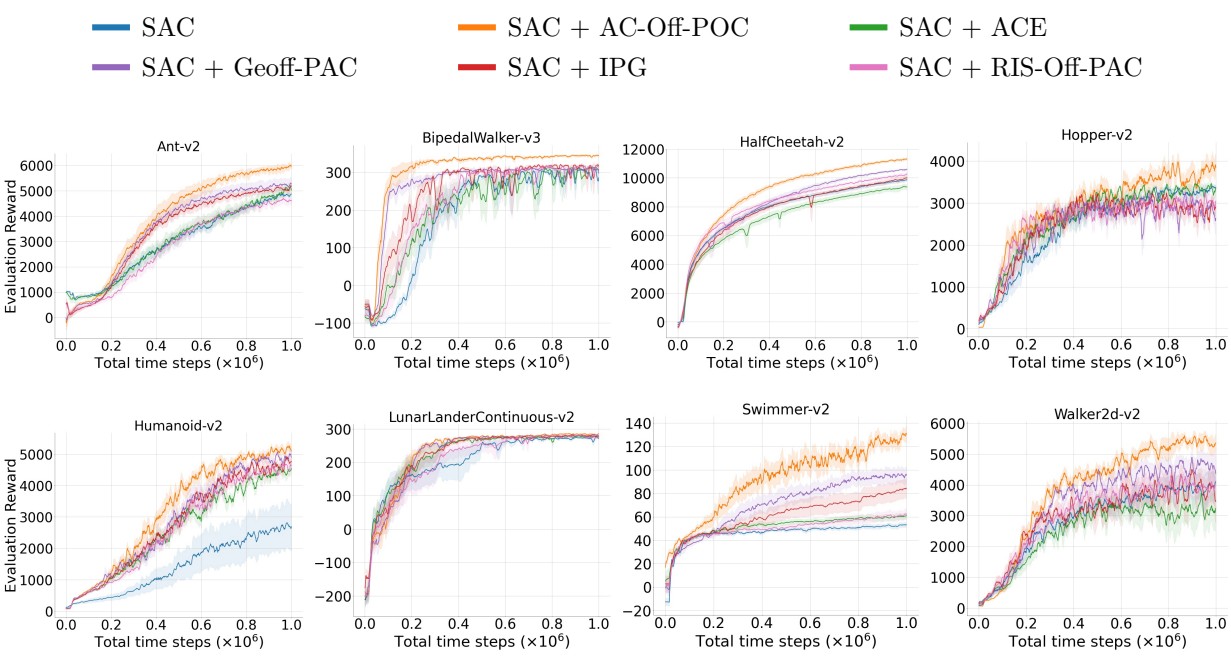

Figure 2: Evaluation curves for the set of OpenAI Gym continuous control tasks for 1 million training time steps over 10 random seeds under the SAC algorithm. The shaded region represents a 95% confidence interval over the trials. A sliding window of size 10 smoothes the curves for visual clarity.

suboptimal during the learning process, which results in below-par baselines. The empirical performance of ACE is confined to simpler domains such as simple Markov chains or cart-pole balancing, as it was initially proposed for linear function approximation. Geoff-PAC, its deep function approximation extension, produces better baseline performance, as expected. However, we posit that the PG estimation of Geoff-PAC, despite its unbiased sample from the policy gradient, is still inaccurate and insufficient for an optimal performance. Lastly, IPG results in an on-policy deterministic actor-critic method when $v = 1$ is employed (Gu et al., 2017). Our earlier discussion proposes that a complete on-policy PG can completely disregard off-policy samples. Nevertheless, off-policy samples can sometimes prove beneficial, depending on the task. Therefore, the maximal performance observed is not by IPG, but rather by AC-Off-POC.

Consequently, off-policy samples usually degrade the performance due to the underlying distribution that substantially diverges from the current agent's policy. Nonetheless, off-policy methods may still require off-policy samples to learn the environment (Cicek et al., 2021). AC-Off-POC solves this issue by employing the Jensen-Shannon divergence, which obtains a smooth similarity measurement prior to the non-linear transformation. Its symmetric similarity measurement prevents the off-policy transitions from being heavily penalized and allows them to contribute to the learning progress even with a small proportion. Considering that the hyperparameters introduced by the competing approaches increase the computational complexity and parameter-free AC-Off-PAC attains superior performance, we believe that our method provides significant gains over the prior work.

## 5.3 Sensitivity Analysis

The batch and replay memory sizes and different experience replay sampling algorithms can considerably impact the learning. Therefore, we perform experiments on both variants of AC-Off-POC under different experience replay buffer and minibatch sizes and sampling algorithms. We evaluate each baseline and AC-Off-POC with minibatch sizes of 32, 256, 1024, and 2048 and the replay memory of sizes 1 million (1M) and 100,000 (100K). We also investigate the performance improvement by AC-Off-POC under the sampling

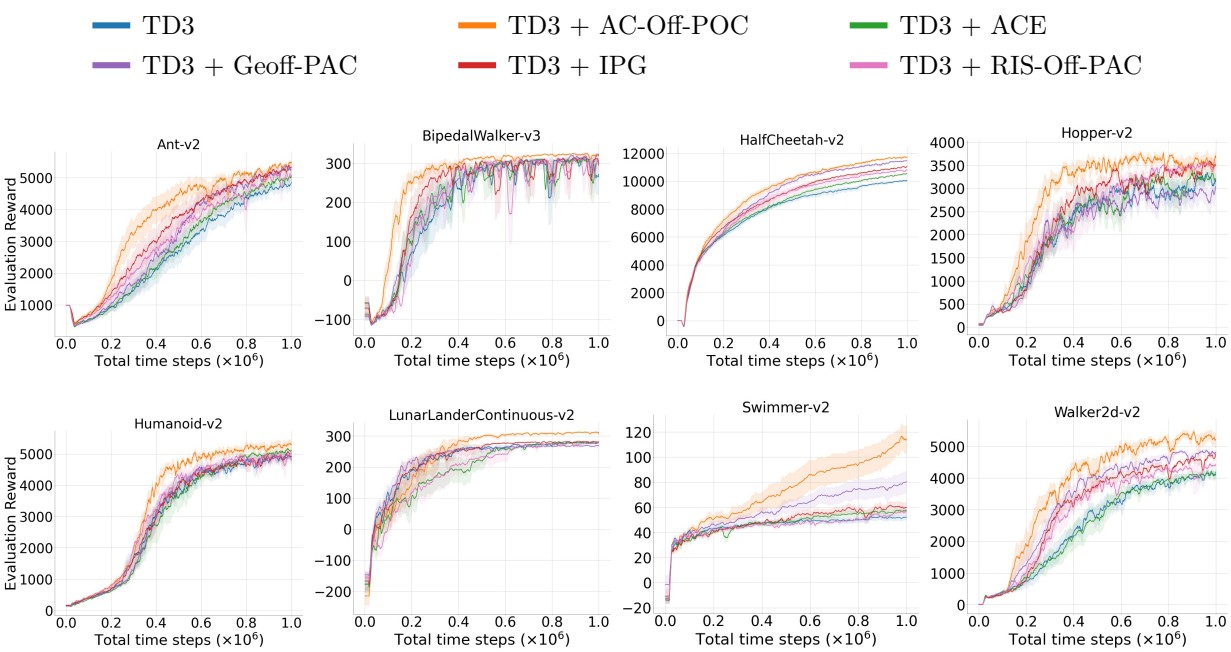

Figure 3: Evaluation curves for the set of OpenAI Gym continuous control tasks for 1 million training time steps over 10 random seeds under the TD3 algorithm. The shaded region represents a 95% confidence interval over the trials. A sliding window of size 10 smoothes the curves for visual clarity.

algorithms of CER, ERO, and PER. The same experimental setting given in Section 5.1 is used. Unless otherwise stated, a minibatch size of 256, replay memory of size 1 million transitions, and uniform sampling are used as the default setting.

### 5.3.1   Impact of the Minibatch Size

Table 4 demonstrates the resulting performances for different batch sizes. Firstly, all methods show optimal performance with a batch size of 256, both in terms of mean and confidence, as they were originally tuned (Haarnoja et al., 2018; Fujimoto et al., 2018). When applying stochastic AC-Off-POC to the SAC algorithm, it is observed that the relative performance improvement brought by AC-Off-POC remains consistent across different batch sizes. This can be attributed to the transition-wise similarity measurement unique to the stochastic variant, meaning that when implemented with stochastic policies, AC-Off-POC is not influenced by minibatch learning and size.

However, the batch-wise similarity measurement in the deterministic variant for TD3 is considerably affected by the minibatch size. For all environments, an increase in batch size negatively impacts the performance of AC-Off-POC and falls short of the baseline performance. Similar performance degradation occurs for smaller batch sizes due to the inaccurate estimation of the JSD between two Gaussians with an insufficient number of samples. In summary, our batch size sensitivity analysis empirically verifies Observation 1.

### 5.3.2   Impact of the Experience Replay Buffer Size

Table 5 shows the effect of varying replay memory sizes, implying substantial shifts in performance in terms of both mean rewards and confidence. In the HalfCheetah environment, decreasing the replay memory size significantly impairs the performance across all methods. This can be attributed to the environment dynamics of HalfCheetah, which typically necessitates off-policy samples for efficient solution. We consider a First-In,

First-Out (FIFO) buffer in all experiments, meaning that a smaller replay memory will contain more on-policy samples compared to a buffer of size 1 million transitions.

In Hopper and Walker2d, a smaller memory size leads to better performance, supporting the evaluation results from Section 5.2. This suggests that off-policy correction in these environments produces more substantial performance enhancement compared to HalfCheetah, as the policy is trained in a more on-policy manner. Similarly, we find that the deterministic variant of AC-Off-POC with a smaller buffer underperforms the baseline in HalfCheetah, but markedly improves the deterministic baseline algorithm in the Hopper and Walker2d environments.

### 5.3.3 Impact of Different Experience Replay Sampling Algorithms

Table 6 reports the resulting performances across different sampling algorithms. We first observe that CER is superior among all the sampling methods. This is due to the most recent collected transition, which is included in each update. Hence, all updates always occur with at least a single on-policy sample. A performance improvement can still be obtained when AC-Off-POC is applied to the baselines while sampling is performed through CER. However, AC-Off-POC slightly underperforms the baselines in the HalfCheetah environment under CER because of the discussed off-policy sample requirement. ERO performs poorly when applied to the baseline algorithms due to the internal structure of the algorithm, which is not in the scope of this study. Nevertheless, AC-Off-POC still improves ERO when applied to the baseline algorithms. In addition, PER usually underperforms when applied to an actor-critic method. This is also an expected result due to the widely known suboptimal empirical performance of PER (Saglam et al., 2022b). Although ERO and PER exhibit poor performance due to the discussed drawbacks, AC-Off-POC can still attain higher rewards. Therefore, we infer that AC-Off-POC can readily adapt to various experience replay sampling algorithms.

Finally, the results obtained under different experience replay sampling schemes support our motivation behind concentrating on one-step returns. Specifically, the recent advances in experience replay research use

Table 2: Average return of last 10 evaluations over 10 trials of 1 million time steps for Ant, BipedalWalker, HalfCheetah, and Hopper. $\pm$ captures a 95% confidence interval over the trials. Bold values represent the maximum under each baseline algorithm and environment.

| Method | Ant | BipedalWalker | HalfCheetah | Hopper |
|---|---|---|---|---|
| DDPG | $816.99 \pm 165.29$ | $-139.73 \pm 36.10$ | $3798.16 \pm 1586.74$ | $2137.20 \pm 537.64$ |
| DDPG + AC-Off-POC | $\mathbf{1052.28 \pm 239.11}$ | $\mathbf{-104.79 \pm 38.39}$ | $\mathbf{5012.32 \pm 2155.64}$ | $\mathbf{2335.00 \pm 765.7}2$ |
| DDPG + ACE | $829.96 \pm 133.42$ | $-122.38 \pm 33.63$ | $4043.05 \pm 1672.13$ | $1742.74 \pm 488.32$ |
| DDPG + Geoff-PAC | $784.67 \pm 133.77$ | $-125.71 \pm 36.54$ | $4660.93 \pm 1951.33$ | $1767.06 \pm 676.08$ |
| DDPG + IPG | $833.51 \pm 88.32$ | $-121.91 \pm 39.11$ | $4248.09 \pm 1796.15$ | $1905.45 \pm 731.71$ |
| DDPG + RIS-Off-PAC | $837.10 \pm 107.24$ | $-107.97 \pm 31.83$ | $4171.61 \pm 1753.42$ | $1819.95 \pm 450.00$ |
| | | | | |
| SAC | $4864.24 \pm 296.94$ | $285.37 \pm 50.74$ | $9871.42 \pm 273.98$ | $3351.43 \pm 317.07$ |
| SAC + AC-Off-POC | $\mathbf{5983.03 \pm 341.04}$ | $\mathbf{343.96 \pm 6.93}$ | $\mathbf{11320.82 \pm 262.23}$ | $\mathbf{3846.05 \pm 777.89}$ |
| SAC + ACE | $5019.17 \pm 549.69$ | $307.18 \pm 17.98$ | $9375.53 \pm 410.10$ | $3367.48 \pm 313.79$ |
| SAC + Geoff-PAC | $5281.92 \pm 471.66$ | $297.24 \pm 29.89$ | $10604.40 \pm 173.26$ | $2636.44 \pm 791.47$ |
| SAC + IPG | $5173.16 \pm 274.65$ | $318.80 \pm 9.41$ | $9993.52 \pm 399.77$ | $2906.38 \pm 609.73$ |
| SAC + RIS-Off-PAC | $4613.55 \pm 430.51$ | $313.67 \pm 12.80$ | $10267.73 \pm 325.33$ | $3036.12 \pm 534.42$ |
| | | | | |
| TD3 | $4846.66 \pm 418.23$ | $271.77 \pm 79.60$ | $10050.95 \pm 168.03$ | $3174.56 \pm 401.05$ |
| TD3 + AC-Off-POC | $\mathbf{5472.19 \pm 276.49}$ | $\mathbf{322.38 \pm 6.92}$ | $\mathbf{11735.35 \pm 196.97}$ | $\mathbf{3658.85 \pm 354.52}$ |
| TD3 + ACE | $5064.34 \pm 376.07$ | $306.96 \pm 15.10$ | $10543.87 \pm 148.98$ | $3215.57 \pm 379.31$ |
| TD3 + Geoff-PAC | $5261.21 \pm 416.70$ | $319.52 \pm 15.19$ | $11467.90 \pm 148.70$ | $3041.78 \pm 516.34$ |
| TD3 + IPG | $5332.47 \pm 367.49$ | $295.09 \pm 44.95$ | $11064.32 \pm 143.90$ | $3524.52 \pm 247.26$ |
| TD3 + RIS-Off-PAC | $5045.43 \pm 488.06$ | $307.22 \pm 16.90$ | $10811.51 \pm 166.86$ | $3532.53 \pm 128.64$ |

Table 3: Average return of last 10 evaluations over 10 trials of 1 million time steps for Humanoid, LunarLanderContinuous, Swimmer, and Walker2d. ± captures a 95% confidence interval over the trials. Bold values represent the maximum under each baseline algorithm and environment.

| Method | Humanoid | LunarLanderContinuous | Swimmer | Walker2d |
|---|---|---|---|---|
| DDPG | 206.80 ± 105.63 | 26.69 ± 73.32 | 19.71 ± 5.90 | 1270.86 ± 332.23 |
| DDPG + AC-Off-POC | **1115.81 ± 641.76** | 173.02 ± 85.59 | **28.95 ± 7.88** | **2854.85 ± 555.45** |
| DDPG + ACE | 204.14 ± 100.09 | 161.61 ± 58.27 | 20.82 ± 12.27 | 1742.84 ± 414.89 |
| DDPG + Geoff-PAC | 255.75 ± 115.61 | **210.53 ± 70.60** | 27.80 ± 5.45 | 2030.25 ± 625.53 |
| DDPG + IPG | 208.46 ± 84.78 | 207.11 ± 69.60 | 24.55 ± 5.47 | 1601.75 ± 516.69 |
| DDPG + RIS-Off-PAC | 198.50 ± 81.86 | 196.59 ± 66.94 | 21.93 ± 6.86 | 1644.48 ± 462.71 |
| | | | | |
| SAC | 2689.22 ± 1499.29 | 271.37 ± 9.82 | 53.11 ± 4.68 | 4103.54 ± 805.03 |
| SAC + AC-Off-POC | **5098.33 ± 529.73** | **284.83 ± 7.09** | **130.29 ± 19.46** | **5366.81 ± 819.00** |
| SAC + ACE | 4545.63 ± 415.48 | 277.18 ± 7.91 | 60.89 ± 5.46 | 3239.37 ± 1193.46 |
| SAC + Geoff-PAC | 4897.81 ± 508.99 | 279.75 ± 9.46 | 96.24 ± 14.80 | 4367.23 ± 1217.94 |
| SAC + IPG | 4768.96 ± 457.70 | 276.29 ± 11.56 | 84.20 ± 16.11 | 3456.34 ± 1486.13 |
| SAC + RIS-Off-PAC | 4646.68 ± 459.60 | 276.06 ± 11.19 | 61.90 ± 9.43 | 4032.21 ± 976.43 |
| | | | | |
| TD3 | 4929.35 ± 331.30 | 277.36 ± 5.42 | 52.18 ± 4.45 | 4107.08 ± 349.81 |
| TD3 + AC-Off-POC | **5297.80 ± 308.16** | **308.05 ± 15.24** | **114.02 ± 21.51** | **5219.89 ± 583.36** |
| TD3 + ACE | 5115.29 ± 219.01 | 279.49 ± 5.27 | 57.18 ± 5.30 | 4179.44 ± 267.09 |
| TD3 + Geoff-PAC | 4805.35 ± 397.55 | 268.35 ± 6.36 | 80.36 ± 16.91 | 4795.22 ± 232.31 |
| TD3 + IPG | 4908.21 ± 482.69 | 280.46 ± 5.32 | 59.94 ± 8.01 | 4751.56 ± 153.17 |
| TD3 + RIS-Off-PAC | 5015.17 ± 340.12 | 278.38 ± 6.68 | 56.18 ± 5.13 | 4421.31 ± 194.74 |

Table 4: Average return over the last 10 evaluations over 10 trials of 1 million time steps, comparing the impact of minibatch sizes $\{32, 256, 1024, 2048\}$. ± captures a 95% confidence interval over the trials. Bold values represent the maximum under each baseline algorithm and environment.

| Setting | HalfCheetah | Hopper | Walker2d |
|---|---|---|---|
| SAC (32) | 9092.21 ± 312.47 | 3304.77 ± 348.16 | 3707.69 ± 844.41 |
| SAC (256) | 9871.42 ± 273.98 | 3351.43 ± 317.07 | 4103.54 ± 805.03 |
| SAC (1024) | 8766.95 ± 290.90 | 2700.17 ± 342.32 | 3848.68 ± 927.21 |
| SAC (2048) | 8477.68 ± 322.43 | 2930.11 ± 324.41 | 3222.36 ± 930.57 |
| SAC + AC-Off-POC (32) | 11022.58 ± 276.83 | 3470.82 ± 777.71 | 5135.93 ± 860.30 |
| SAC + AC-Off-POC (256) | **11320.82 ± 262.23** | **3846.05 ± 777.89** | **5366.81 ± 819.00** |
| SAC + AC-Off-POC (1024) | 10459.58 ± 284.13 | 3269.79 ± 833.98 | 5242.90 ± 823.80 |
| SAC + AC-Off-POC (2048) | 9215.60 ± 303.78 | 3558.28 ± 893.85 | 4300.63 ± 918.97 |
| | | | |
| TD3 (32) | 8955.77 ± 181.85 | 2961.18 ± 434.44 | 3966.54 ± 377.57 |
| TD3 (256) | 10050.95 ± 168.03 | 3174.56 ± 401.05 | 4107.08 ± 349.81 |
| TD3 (1024) | 9963.48 ± 185.11 | 2847.66 ± 408.16 | 3881.20 ± 351.81 |
| TD3 (2048) | 8712.29 ± 194.69 | 2630.64 ± 455.97 | 4033.33 ± 351.51 |
| TD3 + AC-Off-POC (32) | 9104.90 ± 249.96 | 3112.20 ± 411.33 | 4951.56 ± 698.94 |
| TD3 + AC-Off-POC (256) | **11735.35 ± 196.97** | **3658.85 ± 354.52** | **5219.89 ± 583.36** |
| TD3 + AC-Off-POC (1024) | 8791.17 ± 252.00 | 3126.42 ± 426.60 | 5040.96 ± 607.94 |
| TD3 + AC-Off-POC (2048) | 6982.50 ± 248.80 | 2176.53 ± 469.73 | 4713.68 ± 730.17 |

temporally uncorrelated transitions in sampled batches. Although our algorithm can be generalized to the multi-step returns (i.e., trajectories), we focus on one-step algorithms for our method to be compatible with the current experience replay sampling techniques and the ones that will be introduced in the future.

Table 5: Average return over the last 10 evaluations over 10 trials of 1 million time steps, comparing the impact of replay buffer sizes $\{100000, 1000000\}$. $\pm$ captures a 95% confidence interval over the trials. Bold values represent the maximum under each baseline algorithm and environment.

| Setting | HalfCheetah | Hopper | Walker2d |
|---|---|---|---|
| SAC (100K) | $8596.65 \pm 326.68$ | $3446.68 \pm 368.41$ | $4117.61 \pm 937.39$ |
| SAC (1M) | $9871.42 \pm 273.98$ | $3351.43 \pm 317.07$ | $4103.54 \pm 805.03$ |
| SAC + AC-Off-POC (100K) | $8290.28 \pm 309.43$ | $\mathbf{3959.37 \pm 744.74}$ | $\mathbf{5471.21 \pm 788.24}$ |
| SAC + AC-Off-POC (1M) | $\mathbf{11320.82 \pm 262.23}$ | $3846.05 \pm 777.89$ | $5366.81 \pm 819.00$ |
| | | | |
| TD3 (100K) | $9343.67 \pm 188.52$ | $3200.30 \pm 403.49$ | $4183.48 \pm 350.12$ |
| TD3 (1M) | $10050.95 \pm 168.03$ | $3174.56 \pm 401.05$ | $4107.08 \pm 349.81$ |
| TD3 + AC-Off-POC (100K) | $9068.78 \pm 255.80$ | $\mathbf{3790.57 \pm 494.83}$ | $\mathbf{5302.83 \pm 670.45}$ |
| TD3 + AC-Off-POC (1M) | $\mathbf{11735.35 \pm 196.97}$ | $3658.85 \pm 354.52$ | $5219.89 \pm 583.36$ |

Table 6: Average return over the last 10 evaluations over 10 trials of 1 million time steps, comparing the impact of uniform, CER, ERO, and PER sampling methods. $\pm$ captures a 95% confidence interval over the trials. Bold values represent the maximum under each baseline algorithm and environment.

| Setting | HalfCheetah | Hopper | Walker2d |
|---|---|---|---|
| SAC (uniform) | $9871.42 \pm 273.98$ | $3351.43 \pm 317.07$ | $4103.54 \pm 805.03$ |
| SAC (CER) | $10539.72 \pm 285.12$ | $3442.93 \pm 311.60$ | $4105.39 \pm 712.35$ |
| SAC (ERO) | $9497.97 \pm 284.17$ | $3393.43 \pm 315.16$ | $4033.84 \pm 843.72$ |
| SAC (PER) | $8953.30 \pm 296.70$ | $2891.34 \pm 317.16$ | $3844.24 \pm 832.49$ |
| SAC + AC-Off-POC (uniform) | $\mathbf{11320.82 \pm 262.23}$ | $3846.05 \pm 777.89$ | $5366.81 \pm 819.00$ |
| SAC + AC-Off-POC (CER) | $9642.60 \pm 209.10$ | $\mathbf{3905.79 \pm 760.39}$ | $\mathbf{5449.23 \pm 802.84}$ |
| SAC + AC-Off-POC (ERO) | $11006.71 \pm 272.90$ | $3865.37 \pm 765.77$ | $5409.00 \pm 827.40$ |
| SAC + AC-Off-POC (PER) | $10388.39 \pm 297.94$ | $3493.84 \pm 840.43$ | $4402.69 \pm 963.22$ |
| | | | |
| TD3 (uniform) | $10050.95 \pm 168.03$ | $3174.56 \pm 401.05$ | $4107.08 \pm 349.81$ |
| TD3 (CER) | $10628.20 \pm 173.55$ | $3190.81 \pm 400.69$ | $4151.20 \pm 342.95$ |
| TD3 (ERO) | $9591.56 \pm 168.77$ | $3187.00 \pm 403.67$ | $4013.53 \pm 361.32$ |
| TD3 (PER) | $9578.80 \pm 172.61$ | $2637.58 \pm 407.95$ | $3447.57 \pm 350.60$ |
| TD3 + AC-Off-POC (uniform) | $\mathbf{11735.35 \pm 196.97}$ | $3658.85 \pm 354.52$ | $5219.89 \pm 583.36$ |
| TD3 + AC-Off-POC (CER) | $9851.44 \pm 177.24$ | $\mathbf{3692.22 \pm 335.98}$ | $\mathbf{5367.62 \pm 486.86}$ |
| TD3 + AC-Off-POC (ERO) | $11345.45 \pm 205.72$ | $3664.95 \pm 352.06$ | $5039.16 \pm 589.69$ |
| TD3 + AC-Off-POC (PER) | $10648.33 \pm 231.66$ | $3467.80 \pm 415.79$ | $4357.27 \pm 622.60$ |

### 5.3.4 Impact of the Divergence Measure

We test the resulting performance of AC-Off-POC when JSD is replaced with KL divergence. These results are provided in Table 7. We obtain higher results with JSD in all of the environments. However, for the HalfCheetah environment, KL divergence significantly degrades the performance. As it severely penalizes transitions under which the policy distribution differs from the policy of interest, most off-policy transitions are not included in the gradient computation due to the corresponding small weights. As discussed, the HalfCheetah environment usually requires those off-policy transitions, and since the agent does not effectively use them, the performance drops. Overall, these empirical studies also verify our discussions on the advantages of JSD over KL divergence.

Table 7: Average return over the last 10 evaluations over 10 trials of 1 million time steps, comparing the measures of the Jensen-Shannon divergence and KL divergence. $\pm$ captures a 95% confidence interval over the trials. Bold values represent the maximum under each baseline algorithm and environment.

| Setting | HalfCheetah | Hopper | Walker2d |
|---|---|---|---|
| SAC + AC-Off-POC (KL) | $10747.20 \pm 219.16$ | $3567.88 \pm 373.95$ | $4980.37 \pm 620.16$ |
| SAC + AC-Off-POC (JSD) | $\mathbf{11320.82 \pm 262.23}$ | $\mathbf{3846.05 \pm 777.89}$ | $\mathbf{5366.81 \pm 819.00}$ |
| | | | |
| TD3 + AC-Off-POC (KL) | $10112.36 \pm 200.21$ | $3449.99 \pm 362.96$ | $4915.94 \pm 612.65$ |
| TD3 + AC-Off-POC (JSD) | $\mathbf{11735.35 \pm 196.97}$ | $\mathbf{3658.85 \pm 354.52}$ | $\mathbf{5219.89 \pm 583.36}$ |

### 5.4 Weight Analysis

To validate the accuracy of the weights produced by deterministic AC-Off-POC, we conduct experiments employing the experiences of an actor trained under the TD3 algorithm in the HalfCheetah and Hopper tasks, over 1 million time steps. During the training phase, the transitions executed by the agent were stored in temporal order. Then, we start by sampling the transitions with a window size equivalent to the minibatch size centered at each transition. For each sampled batch, the similarity weight $\lambda$ was computed with the deterministic variant of AC-Off-POC with respect to the trained agent's policy.

Figure 4(a) presents the normalized values for time steps and similarity weights produced by AC-Off-POC. From the first transition executed by a random policy to the last transition executed by the expert agent, the sampled batches increasingly become on-policy in relation to the expert agent's policy. It can be observed that as the center of the sliding window nears the last transition, the AC-Off-POC weights approach the value 1. Therefore, the weights rise as the on-policyness of the sampled batch augments, which is observed from the linear trajectory followed by the AC-Off-POC weights. Overall, it can be inferred that the deterministic variant of AC-Off-POC yields accurate estimates with negligible error. However, it does not employ temporally correlated trajectories or any action probability estimate.

We perform the same set of experiments to verify the accuracy of the stochastic variant of AC-Off-POC. As the stochastic variant assigns unique weights to each off-policy sample, we directly depict the weights assigned to each transition contained in the expert agent's experience replay buffer. In Figure 4(b), we observe that the stochastic variant also produces accurate similarity weights to schedule the contribution of each loss

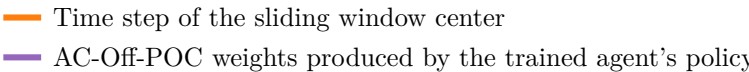

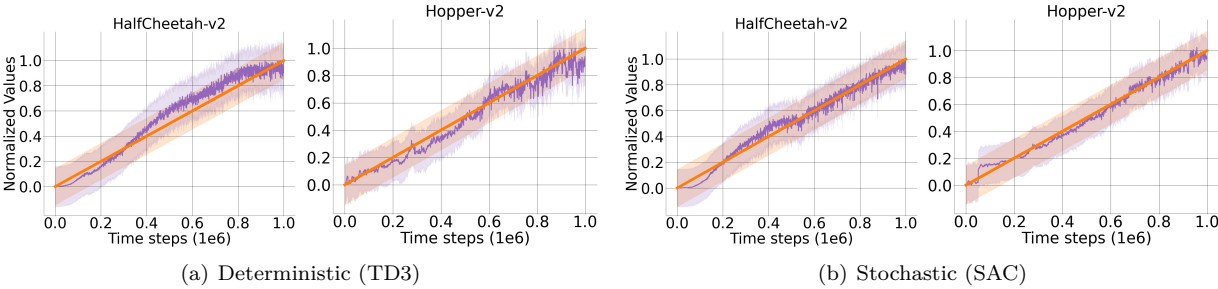

Figure 4: Temporally ordered batches of transitions versus AC-Off-POC weights produced by the trained agent's policy under the TD3 and SAC algorithms for the set of OpenAI Gym continuous control tasks over 1 million training time steps. The shaded region represents a 95% confidence interval of the normalized values over the trials.

component with respect to off-policy samples. Moreover, the confidence of the weights is more robust, and weights are more accurate due to the direct use of the policy distributions in computing the dissimilarity between the target and behavioral policies. In conclusion, as discussed, AC-Off-POC can produce accurate similarity weights by preventing vanishing or exploding gradients and the employment of trajectories due to the transition-wise or temporally uncorrelated batch-wise similarities.

## 6  Future Work

Our technique also has some limitations that open up interesting directions for future work. One limitation is that the choice of Jensen-Shannon divergence can be restrictive as it requires choosing parametric distributions over the action set and storing the distribution parameters along with the $(s, a, r, s')$ tuple in the buffer. This contrasts with the importance weights based on propensity ratio that do not enforce any parametric assumption on the distribution and do not require storing any distribution parameters. A possible extension of our technique is to explore other divergence measures that can relax the parametric assumption and reduce the storage requirement.

Another limitation is that our technique relies on sampling actions from the target policy to estimate the divergence term. This can introduce variance and bias in the estimation, especially when the target policy is very different from the behavioral policy. A possible improvement of our technique is to leverage the entire probability distribution of the target policy instead of sampling actions from it. This can allow us to employ a policy search algorithm to directly learn a policy for solving the underlying MDP of the environment by reducing the variance substantially through multi-importance sampling. Alternatively, if we want to maintain a value function and use its values to learn a policy for a given MDP, i.e., actor-critic methods, then our approach remains a credible off-policy correction technique that also corrects the expected return estimated by the Q-network. We hope that our work will inspire further research on off-policy correction techniques for deep Q-learning variants and their applications to various RL problems.

## 7  Conclusion

We discuss *off-policy correction*, which aims to mitigate the detrimental effects induced by the mismatch between the underlying distributions of the agent's policy and the previously collected data. While off-policy correction is usually performed through off-policy policy gradients (PG) in actor-critic methods, we address that they cannot apply to the policies approximated by the deterministic neural networks since they require action probabilities. Moreover, previous studies usually approach the off-policy correction from the PG side, that is, they adapt the conventional PG techniques that are on-policy by construction to off-policy learning. To this end, we introduce the AC-Off-POC algorithm as a complementary alternative to the prior works. In contrast to off-policy policy gradients, our method approaches the problem from the off-policy data side without modifying the policy gradients used. AC-Off-POC can achieve an efficient one-step off-policy correction and improve the data efficiency by reweighting the contribution of each off-policy sample in optimizing the value function and policy. We support our claims with theoretical analysis to show that it obtains a bounded contraction mapping that offers a safe single-step off-policy correction.

An extensive set of empirical studies demonstrates that AC-Off-POC improves the state-of-the-art and outperforms the competing off-policy correction methods by a considerable margin. By alleviating the bias induced by the off-policy data, our method can attain faster convergence and optimal policies by disregarding the transitions executed by behavioral policies that highly deviate from the current policy in terms of the numerical action decisions. Our method also resolves computational cost issues by not introducing hyperparameters or additional networks. Moreover, we show that a generic approach can readily apply AC-Off-POC to actor-critic method with one-step bootstrapped Q-learning. Lastly, we provide an open-source repository[1] containing all the code and results to further support research on off-policy deep RL.

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

## A   Proof of Lemma 2

**Lemma 8.** *The difference between $\mathcal{H}Q$ and its fixed point $Q^\pi$ is:*

$$\mathcal{H}Q(s,a) - Q^\pi(s,a) = \gamma\mathbb{E}_{s'\sim P_\eta, a'\sim\eta}[\mathbb{E}_\pi(Q-Q^\pi)(s',\cdot) - \lambda'(Q-Q^\pi)(s',a')],$$

*where $\lambda'$ is the similarity coefficient computed for the subsequent transition of $(s,a,r,s')$.*

*Proof.* The proof reduces the proof of Lemma 1 by Munos et al. (2016) to single-step TD learning when off-policy correction by AC-Off-POC is involved. Munos et al. (2016) consider the trajectories starting from $t=0$. Thus, we consider $t=0$ and denote the entities in the next step by superscript, e.g., $a' := a_{t+1}$, since we focus on one-step returns. The general RETRACE operator for multi-step return-based off-policy algorithms is given by

$$\mathcal{R}Q(s,a) := Q(s,a) + \mathbb{E}_\eta\left[\sum_{t\geq 0}\gamma^t(\prod_{i=1}^t c_i)(r_t + \gamma\mathbb{E}_\pi Q(s_{t+1},\cdot) - Q(s_t,a_t))\right], \tag{14}$$

where $c_i$ is the importance weight corresponding to the time step $i$. Setting $t=0$ in the definition of the operator $\mathcal{R}$ yields

$$\begin{aligned}
\mathcal{R}Q(s,a) &= Q(s,a) + \mathbb{E}_\eta\left[\sum_{t\geq 0}\gamma^t(\prod_{i=1}^t c_i)(r_t + \gamma\mathbb{E}_\pi Q(s_{t+1},\cdot) - Q(s_t,a_t))\right]\Bigg|_{t=0} \\
&= Q(s,a) + \mathbb{E}_\eta\left[r + \gamma\mathbb{E}_\pi Q(s',\cdot) - Q(s,a)\right] \\
&= \mathcal{H}Q(s,a),
\end{aligned}$$

where $s := s_0$, $a := a_0$, $r := r_0$, $s' := s_1$, and $\prod_{s=1}^t c_s = 1$ when $t=0$ (Munos et al., 2016). Notice that reduction to single-step off-policy correction yields the AC-Off-POC operator defined in Definition 1. As done by Munos et al. (2016), we rewrite Equation 14:

$$\mathcal{R}Q(s,a) = \sum_{t\geq 0}\gamma^t\mathbb{E}_\eta\left[(\prod_{s=1}^t c_s)(r_t + \gamma[\mathbb{E}_\pi Q(s_{t+1},\cdot) - c_{t+1}Q(s_{t+1},a_{t+1})])\right],$$

which is obtained by bootstrapping. Setting $t = 0$ reduces to the following alternative of $\mathcal{H}$ obtained by bootstrapping:

$$
\begin{aligned}
\mathcal{H}Q(s,a) &= \sum_{t \geq 0} \gamma^t \mathbb{E}_\eta \left[ (\prod_{s=1}^{t} c_s)(r_t + \gamma[\mathbb{E}_\pi Q(s_{t+1}, \cdot) - c_{t+1}Q(s_{t+1}, a_{t+1})]) \right] \Bigg|_{t=0} \\
&= \mathbb{E}_\eta \left[ r + \gamma[\mathbb{E}_\pi Q(s', \cdot) - \lambda' Q(s', a')] \right],
\end{aligned}
$$

where we replace $c_1$ with the next similarity weight $\lambda'$, which is computed for the subsequent transition of $(s, a, r, s')$. The reduction from multi-step IS to one-return off-policy correction theoretically implies that off-policy correction occurs for the expected return of the next state-action pair $(s', a')$ (Harutyunyan et al., 2016). As $Q^\pi$ is the fixed point of $\mathcal{R}$, and hence, $\mathcal{H}$, we have

$$
\mathcal{H}Q^\pi(s,a) = Q^\pi(s,a) = \mathbb{E}_\eta[r + \gamma[\mathbb{E}_\pi Q^\pi(s', \cdot) - \lambda' Q^\pi(s', a')]].
$$

Subtracting the fixed point $Q^\pi(s, a)$ from $\mathcal{H}Q(s, a)$, we get

$$
\begin{aligned}
\mathcal{H}Q(s,a) - Q^\pi(s,a) &= \mathbb{E}_\eta \left[ r - r + \gamma[\mathbb{E}_\pi[Q(s', \cdot) - Q^\pi(s', \cdot)] - \lambda'(Q(s', a') - Q^\pi(s', a'))]] \right] \\
&= \gamma \mathbb{E}_{s' \sim P_\eta, a' \sim \eta} \left[ \mathbb{E}_\pi \Delta Q(s', \cdot) - \lambda' \Delta Q(s', a') \right],
\end{aligned}
$$

where we define $\Delta Q \coloneqq Q - Q^\pi$. $\qquad\square$

