# OpenReview forum: "Mitigating Off-Policy Bias in Actor-Critic Methods with One-Step Q-learning: A Novel Correction Approach"
_TMLR — Accepted by TMLR_

### Review · Reviewer_uJTQ · 2023-11-13

**Summary Of Contributions:**

The paper presents a method to compute importance weights used to train actor and critic networks in off-policy Reinforcement Learning settings, focusing on continuous actions (but with both stochastic and deterministic actors). The core of the contribution is a method to compute similarity measures between an unknown behavior policy and the current actor, for each sample in a minibatch. The interesting aspect of the proposed method is that the similarity measure uses both local (the difference between the action selected by the behavior policy and the actor), and minibatch-global (some computed Gaussian distribution) to produce the weights.

Theoretical and empirical evidence is provided that the proposed method makes sense and leads to good results. The empirical evaluation is particularly thorough and uses state-of-the-art baselines. The results seem very good in every case, illustrating the robustness of the proposed method.

**Audience:**

Yes

**Broader Impact Concerns:**

There does not seem to be any risk of negative broader impacts, and as such no discussion of them seems necessary.

**Claims And Evidence:**

Yes

**Requested Changes:**

There is no critical requested change. I would simply suggest that the paper is slightly reorganized to allow the readers to more easily understand the contribution and its motivation:

- Section 1.2 seems a bit too long and superfluous. It provides an "advertisement" for the proposed method but does not mention the gist of the contribution, the fact that a similarity measure based on $\mathcal{B}_a - \pi(\mathcal{B}_s)$ and a minibatch-specific Gaussian distribution is used. I would suggest to make Section 1.2 smaller, maybe by putting the list of contributions back in Section 1, and replacing Section 1.2 by some sort of table, that succintly compares the hypotheses and properties of AC-Off-POC with families of related work.
- Section 4.1 could be split in two sections. Currently, it both (quite nicely) presents the contribution, equation after equation, but then it spends almost one page justifying the use of the JSD versus other distance metrics. Some other aspects of the contribution, such as convincing the reader that computing per-minibatch state-independent covariance matrices is a good idea, are a bit lacking. When reading that section and Equation 8, I was surprised that the state disappears. The later explanations motivate this approach and somewhat (90%) convince me that the state can safely be ignored, but more arguments would be nice. I would suggest to split the section in 2: first a section that succintly presents the contribution (and the pseudocode of Algorithm 1 at the same time), then a section that motivates every aspect. I don't know in which order the sections should be: either first the motivations and then the algorithm, or first the algorithm and then the motivations.
- Section 4.3.1, "Addressing potential concerns", could have a more informative title. For increased readability, I would suggest to use a Latex "definition" in that section, or subsubsections, to list each concern and then explaining their mitigation.

**Strengths And Weaknesses:**

Strengths:

- The proposed method is easy to implement
- The proposed method makes sense, and the paper is relatively clear
- The results are very encouraging
- The potential impact of this paper is very high. The proposed method improves DDPG, SAC and TD3, 3 SOTA algorithms that are extensively used. The method does so without introducing yet another neural network, or complicated mathematical approaches.

Weaknesses:

- The structure of the paper is sometimes a bit difficult to follow, with the readers often having to keep a queue of questions in their mind, with the questions being answered 1 or 2 pages later. The paper is to me a bit too "top to bottom", first giving results and then only explaining how we got to them.

---

> ### Author Response · Authors · 2023-11-27
> **Response to Reviewer uJTQ**
>
> Thank you very much for your constructive feedback on our paper. We are truly grateful for your acknowledgment of its strengths. Regarding your suggestions on the paper’s structure, we have carefully revised the manuscript accordingly. To assist in reviewing these changes, we have highlighted all modifications in response to your comments in color red.
>
> $\newline$
>
> ### **Presentation in Section 1.2**
>
> In line with your comment, we have revised Section 2.1 for conciseness. Additionally, the revised section now elaborates on the unique similarity measure based on $\mathcal{B}_a - \pi (\mathcal{B}_s)$ and the employment of a minibatch-specific Gaussian distribution. To further illustrate the advantages of our method over previous approaches, we have incorporated a new table (*Table 1 on page 3* in the revised manuscript).
>
> For your convenience, we also provide the updated Section 1.2 (*page 2* in the revised manuscript) here:
>
> > To address the identified limitations, we introduce the Actor-Critic Off-Policy Correction (AC-Off-POC) algorithm. AC-Off-POC offers a novel approach, shifting focus to the employment of previously collected off-policy data. This is achieved by adjusting their contributions to the training process through a stateindependent weighting strategy. Specifically, we construct a multivariate Gaussian distribution based on the numerical deviations between actions chosen by the behavioral policies and the current policy. Each off-policy sample is then weighted according to the divergence from this Gaussian to a standard unit Gaussian. Our empirical results demonstrate that AC-Off-POC significantly enhances the performance of state-of-the-art models in a variety of tasks, outperforming existing off-policy correction methods in challenging continuous control tasks from OpenAI Gym (Brockman et al., 2016). A comparative analysis of the improvements brought by AC-Off-POC over previous studies is presented in Table 1. In the interest of reproducibility, we have made our source code available in our (currently anonymized) GitHub repository$^{1}$.
>
> $\newline$
>
> ### **Structure of Section 4.1**
>
> Following your suggestion, we have split Section 4.1 into: **Section 4.1.1 Methodology** and **Section 4.1.2 Methodological Justifications and Motivations**. We have chosen to start by the algorithm itself since we believe that such a structure would help the readers comprehend the framework more. In the revised manuscript, Section 4.1.1 now solely describes the algorithm, while Section 4.1.2 builds the motivation behind the design choices given in the preceding subsection. Furthermore, we have divided Section 4.1.2 by ``\paragraph{}`` to delve into the details in a more structured manner. These paragraphs are entitled as follows.
> + _The Gaussian assumption_
> + _Using Jensen-Shannon divergence_
> + _Employment of the exponential operator_
> +  _Analysis of the impact of minibatch size_
>
>
> Furthermore, following your suggestion, we have added the following sentence in Section 4.1.1 after Equation 8 (_page 7_ in the revised manuscript), to indicate that the algorithm can be seen as *state-independent*:
>
> > Notice that computation of the mean across numerical action deviations, and consequently the covariance matrix, does not depend directly on state information. Rather, it is _state-independent_. This independence arises because the weights are assigned based on actions resulting from the application of the policy to states within the off-policy data. This approach emphasizes the outcomes of actions rather than the states themselves.
>
> Lastly, we would like to underline that Algorithm 1 cannot be given in Section 4.1 since it also binds the stochastic variant. Nevertheless, we are open to further suggestions to increasing the presentation of our article in this matter.

---

> ### Author Response · Authors · 2023-11-27
> **Response to Reviewer uJTQ (cont.)**
>
> ### **Title of Section 4.3.1**
>
> Thank you for your suggestion. In line with your comment, we have changed the name of Section 4.3.1 to **Potential Failure Cases**. Moreover, since Section 4.3.1 is already a ``\subsubsection{}`` and ``\begin{definition} .. \end{definition}`` is mostly reserved for mathematical definitions, we have opted to examine these two cases under ``\paragraph{}``. For the your convenience, we provide the revised Section 4.3.1 (_pages 10-11_ in the revised manuscript) also here:
>
> > **4.3.1. Potential Failure Cases**
>
> >**Mean difference is zero.** Even when the behavioral policies in a sampled batch are quite different from the current policy, the computed mean difference could be zero. To see this, suppose that the action range is $[-10, 10]$, and the current policy outputs action 0, where half of the sampled past actions are -10, and the other half are 10. The fitted normal distribution will have a zero mean even though the policies differ considerably. In this particular example, the variance in the fitted normal distribution would mitigate this issue, that is, the empirical variance would be greater than the exploration noise added to the actions, and thus, the similarity weights will be small.
>
> >**Different policies selecting the same action.** Relatively different policies might select the same action for a particular state. Yet, their probability distributions may not similar to each other. Then, we would give low importance weights to the corresponding tuple. However, optimization of stochastic policies gradients is not similar to the deterministic policies since the distribution-related parameters are included in the training. For example, in the Soft Actor-Critic (SAC) algorithm (Haarnoja et al., 2018), (log-) probabilities of the actions rather than only the selected actions and the entropy of the policy distributions are also included in calculating the policy gradient. Hence, the distributions themselves should also contribute to computing the similarities. Thus, even if two policies choose the same actions, we give low similarity weights to the corresponding tuple unless their distributions are also similar.

---

### Review · Reviewer_3DcE · 2023-11-14

**Summary Of Contributions:**

The authors aim to mitigate the challenges of off-policy data by including a corrective term for reweighting past data.

This is commonly done with importance weights. In this work, the authors focus on a deep RL setting where the policy is deterministic. This can create some challenges with naive importance weights $\pi(a|s) / \mu(a|s)$ where $\pi$ is a one-hot action.

To get around this, the authors will model all difference between policies as following a multivariate Gaussian. More specifically:

For any minibatch $\mathcal{B}$, we have the set of actions $a \in \mathcal{B}$ from $(s,a)$ examples. For each state $s$ we find $\hat{a} = \pi(s)$, which gives us a batch of action differences $\dot a = a - \hat{a}$, and within each minibatch we can fit a Gaussian to the action diffs, with mean $\mu = 1/N \sum \dot a$ and covariance $\Sigma$.

Assuming that policies are run with exploration noise $\sigma$, as is common in methods like DDPG, if $\mu = \pi$, even for fixed $s$ we will see varying actions in our replay buffer, so the similarity $\rho$ is defined as the Jenson-Shannon divergence $JS(N(\mu, \Sigma) || N(0, \sigma I))$. This makes $\rho = 0$ in the limit of inifnite batch size when $\mu = \pi$. Each transition is then weighted by $\lambda = e^{-rho]$. Since $\rho \ge 0$ this gives a weight of 1 for identical policies that trends to 0 as JS divergence rises.

The authors additionally propose an extension of this idea to stochastic policies (based on saving the distribution parameters of the behavior policy in the replay buffer), but fundamentally it computes similarity + reweights in the same way.

These are used to reweight the actor-critic update. The authors present some results showing this is still a contraction (based on RETRACE's treatment) and show it improves upon DDPG and SAC.

The difference between policies is then defined as the Jenson-Shannon divergence betwee

**Audience:**

Yes

**Claims And Evidence:**

Yes

**Requested Changes:**

No changes requested - I am not a fan of the presentation but I think the paper is presented clearly and am willing to believe this is just up to interpretation of the work.

**Strengths And Weaknesses:**

Importance weights are normally not applicable to deterministic policies without meanningless importance weights of $0$ or $\infty$, so this does present a way to handle that scenario. The weighting is also suprirsingly good on the MuJoCo benchmark. It does require some assumptions of a Gaussian exploration noise (in order to define the base $N(0, \sigma I)$), but this is common enough in continuous control that I don't think it's so bad to add that assumption.

To me the main weakness is that..it doesn't seem very motivated or theoretically valid? The authors mention that the 1-step correction done here avoids issues of vanishing importance weights that occur if you take the product over the trajectory. Normally the reason you take the product over the trajectory is because, from a fixed initial state $s_0$, policies $\pi_1$ and $\pi_2$ will visit subsequent states $s_1$, $s_2$, etc. with different probability, since $\pi_1$'s actions will differ from $\pi_2$. So the most theoretically correct thing to do is to multiply the importance weights over all timesteps, to account for difference in state visitation frequency.

What this work essentially does is say, "state visitation doesn't matter. Just assume it's fine and only handle the action difference."This works out because both DDPG and SAC converge on off-policy data, even without correction, in the tabular case, as long as you eventually visit every $(s,a)$ pair.

So, in short it feels more like an optimization trick that has been dressed up in more theory than it deserves. It's really just showing that you can get faster convergence if you reweight examples by how different their actions are in continuous space. And that's fine, but it overall felt like the paper had a lot of theory that it didn't really need.

---

> ### Author Response · Authors · 2023-11-27
> **Response to Reviewer 3DcE**
>
> Thank you very much for your insightful comments. We appreciate your perspective on the theoretical framing of our work, highlighting its core contribution as an optimization technique that improves convergence speed by reweighting examples in continuous action spaces.
>
> We acknowledge your view that the paper may contain more theoretical content than necessary for presenting the core idea. This feedback is fruitful, offering a perspective on how different aspects of our work are perceived by readers. Balancing theoretical depth with practical demonstration is often subjective, and your comments provide an important viewpoint in this regard. Since no specific changes have been requested, we have not made revisions based on the your comments. We believe maintaining the current level of theoretical detail will benefit readers who value a rigorous theoretical foundation for the optimization technique presented. We remain open to suggestions for future improvements and will consider your feedback as we continue to develop and present our work.
>
> Again, thank you again for your constructive feedback and for acknowledging the clarity of our paper's presentation. Your comments reassure us that the core ideas and contributions of our work have been well received.

---

### Review · Reviewer_3Y3g · 2023-11-22

**Summary Of Contributions:**

The paper proposes a new empirical approach to off-policy RL. They propose new way of weighting actor-critic losses using ideas inspired from off-policy evaluation literature. They show the algorithmic procedure and display performance improvements over baseline such as DDPG over continuous control tasks.

**Audience:**

Yes

**Broader Impact Concerns:**

No.

**Claims And Evidence:**

No

**Requested Changes:**

=== weighted loss vs. off-policy evaluation ===

I can see the motivation of the weighted loss -- weighting the actor-critic losses less if the behavior action is different from target action, and more otherwise. This weighting somehow helps the learning process to focus more on the more on-policy data rather than off-policy data.

However, it is not clear what the actual underlying connection is between the proposed weighting loss and off-policy evaluation method such as Retrace, from which the paper draws inspirations. The connection feels very tangential in that Retrace is provably convergent in the off-policy evaluation and optimization settings, whereas the proposed method here feels more like an empirical trick.

The choice of the Retrace form $\min(\pi/\mu,c)$ is justified by convergence theory, how about here? For example, the decaying form $e^{-\rho}$ feels rather arbitrary and why do we choose exponent as the decreasing function of $\rho$ rather than something else? Do we need an extra coefficient to modulate the decay such that the function is actually $e^{-\alpha\rho}$?

The definition of Gaussian interpolation in Eqn 8 is also a purely empirical approach that does not have any theoretical grounding.

=== analysis ===

The theoretical analysis in 4.3.4 does not feel corresponding to the empirical loss proposed in the paper.

For starters, $HQ$'s behavior policy $\eta$ does not serve much purpose in the expectation that defines the operator because it is an one-step transition. Indeed, fixing $s,a$, the only randomness in the expectation will be the one-step transition $s'\sim p(\cdot|s,a)$. Given a fixed target policy $\pi$, the behavior policy $\eta$ does not play a role in the definition. In Retrace the behavior policy and trace cutting is only meaningful when it is more than 1 step.

Most importantly, the proposed method is **weighting** the loss, rather than **weighting** the back-up target. There is a fundamental difference here. The proposed method weights a mixture of one-step loss functions for critic, which ultimately just carries out one-step bootstrapping. This differs from carrying out a weighted target and regress against it, which leads to the classic Retrace type of approach (and also general dynamic programming to TD-learning kind of approach).

Further I do not see how the theory can meaningfully reduce from stochastic to deterministic policy, since in the latter case the problem of off-policy evaluation with multi-step operator is ill-defined. Indeed, if the target policy is deterministic, it can only be one-step and hence no multi-step learning is involved; if the behavior policy is deterministic, $\pi/\mu$ will blow up and importance sampling is ill-defined in that context.

=== Empirical result ===

The gains are interesting but I think the empirical improvements alone cannot justify the approach being applied. It'd be nice and useful in general to see the sensitivity of the approach to various hyper-parameter settings such as the choice of the decay function $e^{-\rho}$ vs. others, or just simple constant $c$ like in Retrace.

Interestingly, Ant-v2 + DDPG seem to perform poorly and likewise for all the methods considered. This might be an issue with the baseline method itself. In openai spinningup baseline, they show that DDPG works definitely worse on Ant-v2 compared to other more sophisticated methods such as TD3 and SAC [1], I wonder if the authors have any thoughts regarding this -- since the baseline shown in the paper is even worse, its performance keeps dropping since the beginning.

[1] https://spinningup.openai.com/en/latest/spinningup/bench.html

**Strengths And Weaknesses:**

=== Strengths ===

The paper is interesting in that it provides empirical performance over continuous control baselines, using ideas from off-policy learning literature. The empirical improvements seem significant compared to the DDPG baseline.

=== Weaknesses ===

Though the empirical improvements are clear, I think much of the paper can be improved. For example, the theoretical motivation of the paper is unclear -- though the weighting idea is inspired from off-policy learning literature, it is only peripherally related to principled off-policy evaluation literature such as Retrace. It is not clear whether the proposed method converges at all or if converging to the optimal policy. As a result, the proposed method does not have enough theoretical grounding and does not offer much insight into the off-policy learning problem.

---

> ### Author Response · Authors · 2023-11-27
> **Response to Reviewer 3Y3g --- Weighted loss vs. Off-policy evaluation**
>
> Thank you very much for your positive feedback on the empirical performance of our study. Please note that in the revised manuscript, we have highlighted the changes made according to your comments by color blue.
>
> $\newline$
>
> ### **Motivation and connection to RETRACE**
>
> Our motivation for establishing this underlying connection stems from our belief that the framework proposed by Munos et al. [1] offers an elegant method for analyzing off-policy evaluation. This framework can be adapted to a variety of off-policy learning schemes, as demonstrated in our study. We began with the general off-policy evaluation setting described by Munos et al. [1] and narrowed our focus to a specific problem: the application of single-step off-policy correction to a randomly sampled batch of transitions. This focused analysis enabled us to elucidate:
>
> 1. the rationale behind clipping the importance weights computed by AC-Off-POC to $[\frac{\pi(\mathbf{a}{i}|\mathbf{s}{i})}{\eta_{\alpha_{i}}(\mathbf{a}{i}|\mathbf{s}{i})}, e^{-\rho_{i}}]$,
> 2. how AC-Off-POC provides a "safe" off-policy correction, acting as a $\gamma$-contraction mapping in relation to the true Q-function $Q^{\pi}$.
>
> Consequently, we believe that our theoretical analysis is a comprehensive and formal approach to demonstrating both the crucial design choice (1) and the benefit (2) of our method.
>
> Apparently, we missed providing the motivation behind following Munos et al. [1] in the manuscript. Thus, we have added the following short paragraph to the revised manuscript before starting the analysis (_page 12, right under Section 4.3.1_ ):
>
> > In our analysis, we adopt the framework established by Munos et al. (2016). This approach enables us to explain the requirement of clipping the importance weights in AC-Off-POC. Furthermore, by adapting the principles from Munos et al. (2016) to the context of single-step returns on a randomly sampled batch of transitions, we demonstrate that our method functions as a $\gamma$-contraction mapping around the true Q-function $Q^{\pi}$. Such a property ensures that iterative application of the TD operator moves the estimated value function closer to the true value function, i.e., convergence to the true value function.
>
> $\newline$
>
> ### **Choice of the exponential operator $e^{-\rho}$**
>
> Thank you for pointing this out. For your reference, we would like to reiterate the motivation behind our choice of the exponential operator. We selected this operator because it:
>
> + provides a seamless transition from unbounded similarity values to bounded similarity weights, eliminating the need for any quantization process,
> + operates efficiently in constant time for a scalar, specifically, $\mathcal{O}(1)$. Additionally, its practical runtime can be further reduced through the use of SIMD operations found in modern CPUs. In practical terms, this means projecting unbounded similarity values into a bounded interval typically runs in $\mathcal{O}(n)$ for a vector of size $n$. However, thanks to SIMD operations, this can be reduced to $\mathcal{O}(\log(n))$.
>
> These points are elaborated on _pages 8-9_ under the **Employment of the exponential operator.** paragraph of Section 4.1.2 in the revised manuscript.
>
> Further, as you pointed out, there was ambiguity regarding our decision not to use an additional coefficient to modulate decay in the algorithm. One of the key advantages of AC-Off-POC is its avoidance of introducing another hyperparameter. Given that reinforcement learning algorithms are inherently complex and often involve numerous hyperparameters requiring task-specific tuning, we consciously aimed to avoid additional parameters. Hence, we opted not to include an extra coefficient for decay modulation.
>
> We appreciate your attention in highlighting this point, as it could have led to confusion among readers. To address this, we have included the following sentence in the section discussing the benefits of the exponential operator (_page 9, second paragraph_ in the revised manuscript):
>
> > Finally, we chose not to include a coefficient to modulate decay (i.e., $e^{-\alpha\rho}$) to keep the algorithm parameter-free. Our empirical results indicate that using $e^{-\rho}$ already achieves satisfactory performance across various environments. However, it is possible to introduce a task-specific modulation coefficient to accelerate learning, if required.
>
> $\newline$
> $\newline$
>
> #### [1]$\quad$Remi Munos, Tom Stepleton, Anna Harutyunyan, and Marc Bellemare. Safe and efficient off-policy reinforcement learning. In D. Lee, M. Sugiyama, U. Luxburg, I. Guyon, and R. Garnett (eds.), _Advances in Neural Information Processing Systems_, volume 29. Curran Associates, Inc., 2016.

---

> > ### Author Response · Authors · 2023-11-27
> > **Response to Reviewer 3Y3g --- Weighted loss vs. Off-policy evaluation (cont.)**
> >
> > ### **Definition of the Gaussian interpolation in Equation 8**
> >
> > We acknowledge that the Gaussian interpolation technique in Equation 8 is primarily based on empirical evidence. This approach has previously been employed by Cicek et al. [2], demonstrating its effectiveness in empirical results. To underscore this point, we have cited the study of Cicek et al. [2], right after Equation 8 in the revised manuscript (_page 7_). The corresponding sentence is as follows:
> >
> > > As an empirical validation, the Gaussian interpolation outlined in the latter equation has been previously employed by Cicek et al. (2021), where it has been proven to enhance overall empirical performance.
> >
> > $\newline$
> > $\newline$
> >
> > #### [2]$\quad$Dogan C. Cicek, Enes Duran, Baturay Saglam, Furkan B. Mutlu, and Suleyman S. Kozat. Off-policy correction for deep deterministic policy gradient algorithms via batch prioritized experience replay. In _2021 IEEE 33rd International Conference on Tools with Artificial Intelligence (ICTAI)_, pp. 1255–1262, 2021. doi: [10.1109/ICTAI52525.2021.00199](https://ieeexplore.ieee.org/document/9643162).

---

> ### Author Response · Authors · 2023-11-27
> **Response to Reviewer 3Y3g --- Analysis**
>
> ### **Operator $HQ$**
>
> In the first draft, we did not explicitly state that our definition of the Bellman residual of $Q$ for policy $\pi$ differs from that of Munos et al. [1]. While Munos et al. defines the residual over a single trajectory, our definition applies it across an average over a dataset, such as sampled batches of transitions or experience replay. This approach is particularly relevant for our theoretical analysis, where we apply it to batches of transitions. For instance, this definition is used in the first equation of the proof of Theorem 3 (_first equation on page 13_ in the revised manuscript). We have added the following clarification to **Definition 1** in the revised manuscript (_page 12_).
>
> > $\dots$ where we define $\mathbb{E}\_{\eta}[\cdot] \coloneqq \mathbb{E}\_{s \sim P_{\eta}, a \sim \eta, r, s^{\prime} \sim P}[\cdot]$ and $\mathbb{E}\_{\pi}Q(s, \cdot) \coloneqq \sum\_{a}\pi(a | s)Q(s, a)$, taken over the dataset of interest containing the transitions.
>
> $\newline$
>
> ### **Weighting in our method vs. RETRACE**
>
> Thank you for this insightful comment. It is indeed accurate that our proposed method diverges from the approach of Munos et al. [1] by specifically weighting the loss rather than the back-up target. The core aim of our theoretical analysis is to highlight this distinction. By starting from a broad formulation of off-policy evaluation, we progressively narrow down to the specific scenario relevant to our study: one-step bootstrapping over a batch of data. This step-by-step approach in our analysis is designed to help readers clearly understand how our proposed method sets itself apart from other methodologies documented in the literature.
>
> $\newline$
>
> ### **Reduction from stochastic to deterministic policies**
>
> We wish to clarify that our theoretical analysis is specifically tailored for one-step bootstrapped Q-learning. In this framework, the issue of ill-definition concerning deterministic policies is effectively resolved. The critical aspect of our analysis, particularly in Theorem 3, involves the action probabilities, namely $\pi(\cdot | s)$ and $\eta(\cdot | s)$. We employ Lemma 2 to demonstrate that
> $$
> \mathcal{H}Q(s, a) - Q^{\pi}(s, a) = \gamma \mathbb{E}{s^{\prime} \sim P{\eta}, \mathcal{B}^{\prime} \sim \mathcal{R}}[\sum_{y}(\pi(y|s^{\prime}) - \eta(y|s^{\prime})\Lambda(y, \mathcal{B}^{\prime}))\Delta Q(s^{\prime}, y)].
> $$
> Here, the condition $\pi(y|s^{\prime}) - \eta(y|s^{\prime})\Lambda(y, \mathcal{B}) \geq 0$ is necessary as it represents a probability. To maintain this, we ensure (and accordingly clip) that $\Lambda(y, \mathcal{B}^{\prime}) \leq \frac{\pi(y|s^{\prime})}{\eta(y|s^{\prime})}$. In this context, the formulation remains well-defined for stochastic policies. Transitioning to deterministic policies, the key difference is the probabilities taking values of either 1 or 0, with $y$ being the deterministic action chosen by the target policy $\pi$. This scenario remains well-defined, as clipping is not applied in deterministic cases, avoiding zero probabilities in the denominator.
>
> We acknowledge and emphasize that our problem formulation is specifically valid in this context. As you observed, any deviation from this setup could render the problem ill-defined for deterministic policies. We appreciate this observation and are open to further elaboration should there be any additional questions or concerns.

---

> ### Author Response · Authors · 2023-11-27
> **Response to Reviewer 3Y3g --- Empirical Result**
>
> ### **Potential comparison of the decay function $e^{-\rho}$ vs. others**
>
> We agree that incorporating comparisons with different decay functions could significantly enhance the value of our study. However, the current scope of empirical research already comprises a substantial portion of the article, and expanding it further, particularly to include such comparisons, might detract from its current focus. We truly appreciate this insightful suggestion, and it certainly presents a compelling direction for future research.
>
> Moreover, given our current resource constraints and the limited time frame of two weeks for completing this revision, it is unfortunately not feasible for us to undertake additional comparative experiments at this stage. We hope to explore these aspects in subsequent studies and believe that this would be a valuable extension of our current work.
>
> $\newline$
>
> ### **DDPG in the Ant-v2 environment**
>
> The observed suboptimal performance in the Ant environment can be primarily attributed to the baseline method itself. It is important to note that in the Ant environment, episode rewards around 1000 are generally indicators of random behavior. In fact, a randomly initialized policy can achieve episode rewards in the range of 800-900. Therefore, we did not observe any significant performance improvement in the Ant environment with any of the algorithms we tested. While ours could improve the performance more than the competing methods, it is still not notable compared to what has been achieved in the SAC and TD3 experiments.
>
> The crux of this issue lies with the DDPG algorithm. To illustrate, consider the comparison between DDPG and TD3. TD3 enhances DDPG by making minor modifications (with major effects) to the Q-learning mechanism, specifically to reduce the overestimation of Q-values. Despite the simplicity of these changes (which could amount to as little as 2-3 lines of code), they result in a substantial improvement in performance. This suggests that the poor performance of DDPG in the Ant environment is not primarily due to its off-policy learning component. Instead, it is the overestimation of Q-values within DDPG that hinders its ability to improve over time in this specific setting.
>
> In light of this, the performance curves for the Ant environment presented in our article merely reflect random behaviors. If these results were scaled to the maximum reward ever achieved by a reinforcement learning algorithm in this environment (approximately 6000, as achieved by SAC according to [3]), the differences between the curves would be practically the same. Thus, it is reasonable to conclude that the results reported for DDPG in our study, as well as those in OpenAI's baselines and other literature, predominantly represent random policy behaviors.
>
> $\newline$
> $\newline$
>
> #### [3]$\quad$**OpenAI** Spinning Up [https://spinningup.openai.com/en/latest/spinningup/bench.html](https://spinningup.openai.com/en/latest/spinningup/bench.html)

---

### Author Response · Authors · 2023-11-27
**High-Level Comment**

We would like to thank the reviewers for their thorough reading of our paper and valuable comments. In the revised version of the article, and in the responses that follow, we have addressed the issues highlighted by the reviewers. Below, we provide a point-by-point response to the reviewers' comments and explain how these have been addressed in the revised paper.

We would like to highlight that the technique proposed in our study is an empirical method designed to enhance the performance of off-policy actor-critic algorithms. Besides the comprehensive simulations reported in the article, our mathematical analysis aims to:
   + offer theoretical support where appropriate,
   + provide guarantees where possible, such as convergence (i.e., $\gamma$-contraction mapping),
   + illustrate how the proposed method can be theoretically conceptualized by reducing from the general off-policy evaluation framework.

Furthermore, the design choices in this study are grounded in theoretical understanding and insights from previous research in the field. We have endeavored to present these in a clear and succinct manner.

Please note that the revisions made to the first draft are summarized above, under **Changes Since Last Submission**.

---

### Author Response · Authors · 2024-01-29
**Post Acceptance Comment**

We would like to thank the associate editor and reviewers for their constructive feedback and for recognizing the improvements in our manuscript. We appreciate the detailed comments which guided our revisions and are grateful for the acceptance of our paper. We're pleased that our rebuttal clarified the concerns raised and look forward to contributing to the field.

---

### Decision · Action_Editor_6gDk · 2024-01-23

**Recommendation:** Accept as is

**Comment:**

## Summary

This paper proposes an approach to mitigate off-policy bias in actor-critic reinforcement learning algorithms. Off-policy bias arises when the data used to train the agent's policy differs from the policy it is intended to optimize. This can lead to instability and poor performance. The proposed method uses a novel policy similarity measure to identify and correct off-policy data. This method applies to deterministic policy networks and can be used with one-step Q-learning. The authors show the proposed approach improves the performance over state-of-the-art methods on various continuous control tasks.

## Decision

Overall, the method is novel, and the reviewers found it novel. The initial version of this paper was badly written, and the authors carefully rewrote the paper based on the reviews submitted and significantly improved it. Now, the current version of the paper paper reads well. As a result, the majority of the reviewers lean towards accepting. However, reviewer 3Y3g pointed out the paper is built upon somewhat shaky theoretical foundations because the theory of importance sampling for off-policy correction does not apply to the deterministic policy. However, I think the authors clarified that in the rebuttal.

**Audience:**

I think this would be interesting for TMLR's RL-leaning audience. In particular, considering that the off-policy RL is becoming more and more prominent and essential in several applications, the findings of this paper would be valuable for the RL community.

**Claims And Evidence:**

The authors have provided empirical results on OpenAI gym environments. Their empirical results are convincing and improve significantly than the baselines. The baselines are fair, and the experimental protocol is well-explained. The authors also provide some theory to justify the approach, but I would say the theory side of the paper is not as satisfying or interesting as the experimental results reported.